# Lewis's Signaling Game as beta-VAE For Natural Word Lengths and Segments

**Ryo Ueda**
The University of Tokyo
ryoryoueda@is.s.u-tokyo.ac.jp

**Tadahiro Taniguchi**
Kyoto University    Ritsumeikan University
taniguchi@ci.ritsumei.ac.jp

## Abstract

As a sub-discipline of evolutionary and computational linguistics, emergent communication (EC) studies communication protocols, called emergent languages, arising in simulations where agents communicate. A key goal of EC is to give rise to languages that share statistical properties with natural languages. In this paper, we reinterpret Lewis's signaling game, a frequently used setting in EC, as beta-VAE and reformulate its objective function as ELBO. Consequently, we clarify the existence of prior distributions of emergent languages and show that the choice of the priors can influence their statistical properties. Specifically, we address the properties of word lengths and segmentation, known as Zipf's law of abbreviation (ZLA) and Harris's articulation scheme (HAS), respectively. It has been reported that the emergent languages do not follow them when using the conventional objective. We experimentally demonstrate that by selecting an appropriate prior distribution, more natural segments emerge, while suggesting that the conventional one prevents the languages from following ZLA and HAS.

## 1 Introduction

Understanding how language and communication emerge is one of the ultimate goals in evolutionary linguistics and computational linguistics. *Emergent communication* (EC, Lazaridou & Baroni, 2020; Galke et al., 2022; Brandizzi, 2023) attempts to simulate the emergence of language using techniques such as deep learning and reinforcement learning. A key challenge in EC is to elucidate how emergent communication protocols, or *emergent languages*, reproduce statistical properties similar to natural languages, such as compositionality (Kottur et al., 2017; Chaabouni et al., 2020; Ren et al., 2020), word length distribution (Chaabouni et al., 2019; Rita et al., 2020), and word segmentation (Resnick et al., 2020; Ueda et al., 2023). In this paper, we reformulate *Lewis's signaling game* (Lewis, 1969; Skyrms, 2010), a popular setting in EC, as beta-VAE (Kingma & Welling, 2014; Higgins et al., 2017) for the emergence of more natural word lengths and segments.

**Lewis's signaling game and its objective:** Lewis's signaling game is frequently adopted in EC (e.g., Chaabouni et al., 2019; Rita et al., 2022b). It is a simple communication game involving a *sender* $S_\phi$ and *receiver* $R_\theta$, with only unidirectional communication allowed from $S_\phi$ to $R_\theta$. In each play, $S_\phi$ obtains an *object* $x$ and converts it into a *message* $m$. Then, $R_\theta$ interprets $m$ to guess the original object $x$. The game is successful upon a correct guess. $S_\phi$ and $R_\theta$ are typically represented as neural networks such as RNNs (Hochreiter & Schmidhuber, 1997; Cho et al., 2014) and optimized for a certain objective $\mathcal{J}_{\text{conv}}$, which is in most cases defined as the maximization of the log-likelihood of the receiver's guess (or equivalently minimization of cross-entropy loss).

**ELBO as objective:** In contrast, we propose to regard the signaling game as (beta-)VAE (Kingma & Welling, 2014; Higgins et al., 2017) and utilize ELBO (with a KL-weighing parameter $\beta$) as the game objective. We reinterpret the game as representation learning in a generative model, regarding $x$ as an observed variable, $m$ as latent variable, $S_\phi$ as encoder, $R_\theta$ as decoder, and an additionally introduced "language model" $P_\theta^{\text{prior}}$ as prior latent distribution. Several reasons support our reformulation. First, the conventional objective $\mathcal{J}_{\text{conv}}$ actually resembles ELBO in some respects. We illustrate their similarity in Figure 1 and Table 1. $\mathcal{J}_{\text{conv}}$ has an *implicit prior message distribution*, while ELBO contains explicit one. Also, $\mathcal{J}_{\text{conv}}$ is often adjusted with an *entropy regularizer* while ELBO contains an *entropy maximizer*. Although they are not mathematically equal, their motiva-

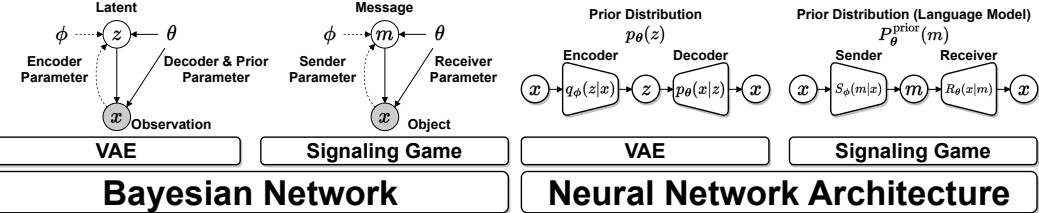

Figure 1: Illustration of similarity between signaling games and (beta-)VAE.

Table 1: A list of objective functions in this paper. It is useful to grasp their similarity.

| objective | definition | prior (implicit/explicit) | | entropy-related term |
|---|---|---|---|---|
| $\mathcal{J}_{\text{conv}}(\boldsymbol{\phi}, \boldsymbol{\theta})$ | Eq. 2 (Section 2) | $P^{\text{prior}}_{\text{unif}}(M)$ | (implicit) | entropy regularizer |
| $\mathcal{J}_{\text{conv}}(\boldsymbol{\phi}, \boldsymbol{\theta}; \alpha)$ | Eq. 4 (Section 2) | $P^{\text{prior}}_{\alpha}(M)$ | (implicit) | entropy regularizer |
| $\mathcal{J}_{\text{elbo}}(\boldsymbol{\phi}, \boldsymbol{\theta}; \alpha)$ | Eq. 12 (Section 3) | $P^{\text{prior}}_{\alpha}(M)$ | (explicit) | entropy maximizer |
| $\mathcal{J}_{\text{ours}}(\boldsymbol{\phi}, \boldsymbol{\theta}; \beta)$ | Eq. 7 (Section 3) | $P^{\text{prior}}_{\boldsymbol{\theta}}(M)$ | (explicit) | entropy maximizer |

tions coincide in that they keep the sender's entropy high to encourage exploration (Levine, 2018). Second, the choice of a prior message distribution can affect the statistical properties of emergent languages. Specifically, this paper addresses *Zipf's law of abbreviation* (ZLA, Zipf, 1935; 1949) and *Harris's articulation scheme* (HAS, Harris, 1955; Tanaka-Ishii, 2021). It has been reported that emergent languages do not give rise to these properties as long as the conventional objective $\mathcal{J}_{\text{conv}}$ is used (Chaabouni et al., 2019; Ueda et al., 2023). We suggest that such an artifact is caused by unnatural prior distributions and that it has been obscure due to their implicitness. Moreover, our ELBO-based formulation can be justified from the computational psycholinguistic point of view, namely *surprisal theory* (Hale, 2001; Levy, 2008; Smith & Levy, 2013). Interestingly, the prior distribution can be seen as a language model, and thus the corresponding term can be seen as the *surprisal*, or cognitive processing cost, of sentences (messages). In that sense, ELBO naturally models the trade-off between the informativeness and processability of sentences.

**Related work and contribution:** The idea *per se* of viewing communication as representation learning is also seen in previous work. For instance, the linguistic concept *compositionality* is often regarded as *disentanglement* (Andreas, 2019; Chaabouni et al., 2020; Resnick et al., 2020).Also, Tucker et al. (2022) formulated a *VQ-VIB* game based on *Variational Information Bottleneck* (VIB, Alemi et al., 2017), which is known as a generalization of beta-VAE. Moreover, Taniguchi et al. (2022) defined a *Metropolis-Hastings (MH) naming game* in which communication was formulated as representation learning on a generative model and emerged through MCMC. However, to the best of our knowledge, no previous work has addressed in an integrated way (1) the similarities between the game and (beta-)VAE from the generative viewpoint, (2) the potential impact of prior distribution on the properties of emergent languages, and (3) a psycholinguistic reinterpretation of emergent communication. They are the main contribution of this paper. Another important point is that, since Chaabouni et al. (2019), most EC work on signaling games has adopted either the one-character or fixed-length message setting, avoiding the variable-length setting, though the variable-length is more natural. We revisit the variable-length setting towards a better simulation of language emergence.

## 2 BACKGROUND

**Mathematical Notation:** Throughout this paper, we use a calligraphic letter for a set (e.g., $\mathcal{X}$), an uppercase letter for a random variable of the set (e.g., $X$), and a lowercase letter for an element of the set (e.g., $x$). We denote a finite object space by $\mathcal{X}$, a finite alphabet by $\mathcal{A}$, a finite message space by $\mathcal{M}$, a probability distribution of objects by $P_{\text{obj}}(X)$, a probabilistic sender agent by $S_{\boldsymbol{\phi}}(M|X)$, and a probabilistic receiver agent by $R_{\boldsymbol{\theta}}(M|X)$. $S_{\boldsymbol{\phi}}$ is parametrized by $\boldsymbol{\phi}$ and $R_{\boldsymbol{\theta}}$ by $\boldsymbol{\theta}$.

### 2.1 LANGUAGE EMERGENCE VIA LEWIS'S SIGNALING GAME

To simulate language emergence, defining an environment surrounding agents is necessary. One of the most standard settings in EC is *Lewis's signaling game* (Lewis, 1969; Skyrms, 2010; Rita et al., 2022b), though there are variations in the environment definition (Foerster et al., 2016; Lowe et al., 2017; Mordatch & Abbeel, 2018; Jaques et al., 2019, *inter alia*).[1] The game involves two agents,

---

[1]Strictly speaking, our subject is the *reconstruction* game, though the *discrimination* (*referential*) game is also frequently studied (Lazaridou et al., 2017; Choi et al., 2018; Dessì et al., 2021; Ri et al., 2023).

sender $S_\phi$ and receiver $R_\theta$. In a single play, $S_\phi$ observes an object $x$ and generates a message $m$. $R_\theta$ obtains the message $m$ and guesses the object $x$. The game is successful if the guess is correct. $S_\phi$ and $R_\theta$ are often represented as RNNs and optimized toward successful communication. During training, the sender probabilistically observes an object $x \sim P_{\text{obj}}(\cdot)$, generates a message $m \sim S_\phi(\cdot|x)$, and the receiver guesses via $\log R_\theta(x|m)$. During validation, the sender observes $x$ one by one and greedily generates $m$.

**Object Space:** Objects $x$ can be defined in various ways, from abstract to realistic data. In this paper, an object $x$ is assumed to be an *attribute-value (att-val) object* (Kottur et al., 2017; Chaabouni et al., 2020) which is a $n_{\text{att}}$-tuple of integers $(v_1, \ldots, v_{n_{\text{att}}}) \in \mathcal{X}$, where $v_i \in \{1, \ldots, n_{\text{val}}\}$ for each $i$. $n_{\text{att}}$ is called the *number of attributes* and $n_{\text{val}}$ is called the *number of values*.

**Message Space:** In most cases, the message space $\mathcal{M}$ is defined as a set of sequences of a (maximum) length $L_{\max}$ over a finite alphabet $\mathcal{A}$. The message length, denoted by $|m|$, can be either fixed (Chaabouni et al., 2020) or variable (Chaabouni et al., 2019). This paper adopts the latter setting:

$$\mathcal{M} := \{m_1 \cdots m_k \mid m_i \in \mathcal{A}\backslash\{\texttt{eos}\}\ (1 \le i \le k-1), m_k = \texttt{eos}, 1 \le k \le L_{\max}\}, \quad (1)$$

where $\texttt{eos} \in \mathcal{A}$ is the end-of-sequences marker. We assume $|\mathcal{A}| \ge 3$ and thus $\log(|\mathcal{A}| - 1) > 0$.

**Game Objective:** The game objective is often defined as (Chaabouni et al., 2019; Rita et al., 2022b):

$$\mathcal{J}_{\text{conv}}(\phi, \theta) := \mathbb{E}_{x \sim P_{\text{obj}}(\cdot), m \sim S_\phi(\cdot|x)}\left[\log R_\theta(x \mid m)\right]. \quad (2)$$

For clarity, we refer to $\mathcal{J}_{\text{conv}}$ as the *conventional* objective, in contrast to *our* objective $\mathcal{J}_{\text{ours}}$ defined later. The gradient of $\mathcal{J}_{\text{conv}}(\phi, \theta)$ is obtained as (Chaabouni et al., 2019):

$$\nabla_{\phi,\theta}\mathcal{J}_{\text{conv}}(\phi,\theta) = \mathbb{E}_{x \sim P_{\text{obj}}(\cdot), m \sim S_\phi(\cdot|x)}[\nabla_{\phi,\theta}\log R_\theta(x|m) + (\log R_\theta(x|m) - B(x))\nabla_{\phi,\theta}\log S_\phi(m|x)],$$

where $B : \mathcal{X} \to \mathbb{R}$ is a *baseline*. Intuitively, $\phi$ is updated via REINFORCE (Williams, 1992) and $\theta$ via the standard backpropagation. In practice, an *entropy regularizer* (Williams & Peng, 1991; Mnih et al., 2016) is added to $\nabla_{\phi,\theta}\mathcal{J}_{\text{conv}}(\phi, \theta)$:

$$\nabla_{\phi,\theta}^{\text{entreg}} := \frac{\lambda^{\text{entreg}}}{|m|} \sum_{t=1}^{|m|} \nabla_{\phi,\theta}\,\mathcal{H}(S_\phi(M_t \mid x, m_{1:t-1})), \quad (3)$$

where $\lambda^{\text{entreg}}$ is a hyperparameter adjusting the weight. It encourages the sender agent to explore the message space during training by keeping its policy entropy high. *A message-length penalty* $-\alpha|m|$ is sometimes added to the objective to prevent messages from becoming unnaturally long:

$$\mathcal{J}_{\text{conv}}(\phi, \theta; \alpha) := \mathbb{E}_{x \sim P_{\text{obj}}(\cdot), m \sim S_\phi(\cdot|x)}\left[\log R_\theta(x \mid m) - \alpha|m|\right]. \quad (4)$$

Chaabouni et al. (2019) and Rita et al. (2020) pointed out that the message-length penalty is necessary to give rise to Zipf's law of abbreviation (ZLA) in emergent languages.

## 2.2 ON THE STATISTICAL PROPERTIES OF LANGUAGES

It is a key challenge in EC to fill the gap between emergent and natural languages. Though it is not straightforward to evaluate the emergent languages that are uninterpretable for human evaluators, their natural language likeness has been assessed indirectly by focusing on their statistical properties (e.g., Chaabouni et al., 2019; Kharitonov et al., 2020). We briefly introduce previous EC studies on Zipf's law of abbreviation and Harris's articulation scheme.

**Word Lengths:** *Zipf's law of abbreviation* (ZLA, Zipf, 1935; 1949) refers to the statistical property of natural language where frequently used words tend to be shorter. Chaabouni et al. (2019) reported that emergent languages do not follow ZLA by the conventional objective $\mathcal{J}_{\text{conv}}(\phi, \theta)$. They defined $P_{\text{obj}}(X)$ as a power-law distribution. According to ZLA, it was expected that $|m^{(1)}| < |m^{(2)}|$ overall for $x^{(1)}, x^{(2)}$ such that $P_{\text{obj}}(x^{(1)}) > P_{\text{obj}}(x^{(2)})$. Their experiment, however, showed the opposite tendency $|m^{(1)}| > |m^{(2)}|$. They also reported that the emergent languages follow ZLA when the message-length penalty (Eq. 4) is additionally introduced. Previous work has ascribed such anti-efficiency to the lack of desirable inductive biases, such as the sender's *laziness* (Chaabouni et al., 2019) and receiver's *impatience* (Rita et al., 2020), and *memory constraints* (Ueda & Washio, 2021).

**Word Segments:** *Harris's articulation scheme* (HAS, Harris, 1955; Tanaka-Ishii, 2021) is a statistical property of word segmentation in natural languages. According to HAS, the boundaries of word segments in a character sequence can be predicted to some extent from the statistical behavior

of character $n$-grams. This scheme is based on the discovery of Harris (1955) that there tend to be word boundaries at points where the number of possible successive characters after given contexts increases in English corpora. HAS reformulates it based on information theory (Cover & Thomas, 2006). Formally, let BE($s$) be the *branching entropy* of character sequences $s = a_1 \cdots a_n$ ($a_i \in \mathcal{A}$):

$$\mathrm{BE}(s) := \mathcal{H}(A_{n+1}|A_{1:n}=s) = -\sum_{a \in \mathcal{A}} P(A_{n+1}=a|A_{1:n}=s) \log_2 P(A_{n+1}=a|A_{1:n}=s), \quad (5)$$

where $n$ is the length of $s$ and $P(A_{n+1} = a|A_{1:n} = s)$ is defined as a simple $n$-gram model. BE($s$) is known to decrease monotonically on average (Bell et al., 1990), while its increase can be of special interest. HAS states that there tend to be word boundaries at BE's increasing points. HAS can be utilized as an unsupervised word segmentation algorithm (Tanaka-Ishii, 2005; Tanaka-Ishii & Ishii, 2007) whose pseudo-code is shown in Appendix A. However, whether obtained segments from emergent languages are meaningful is not obvious. For instance, no one would believe that segments obtained from a random unigram model represent meaningful words; emergent languages face a similar issue as they are not human languages. We do not even have a direct method to evaluate the meaningfulness since there is no ground-truth segmentation data in emergent languages. To alleviate it, Ueda et al. (2023) proposed the following criteria to indirectly verify their meaningfulness:

C1. The number of boundaries per message (denoted by $n_{\mathrm{bou}}$) should increase if $n_{\mathrm{att}}$ increases.

C2. The number of distinct segments in a language (denoted by $n_{\mathrm{seg}}$) should increase if $n_{\mathrm{val}}$ increases.

C3. W-TopSim > C-TopSim should hold, or $\Delta_{\mathrm{w,c}} :=$ W-TopSim $-$ C-TopSim $> 0$ should hold.[2]

These are rephrased to some extent. C-TopSim is the normal TopSim that regards characters $a \in \mathcal{A}$ as symbol units, whereas W-TopSim regards segments (chunked characters) as symbol units. In the att-val setting, meaningful segments are expected to represent attributes and values in a disentangled way, motivating C1 and C2. On the other hand, C3 is based on the concept called *double articulation* by Martinet (1960), who pointed out that, in natural language, characters within a word are hardly related to the corresponding meaning whereas the word is associated. For instance, characters `a`, `e`, `l`, `p` are irrelevant to a rounded, red fruit, but word `apple` is relevant to it. Thus, the word-level compositionality is expected to be higher than the character-level one, motivating C3. With the conventional objective, however, emergent languages did not satisfy any of them, and the potential causes for this have not been addressed well in Ueda et al. (2023).

## 3 REDEFINE OBJECTIVE AS ELBO FROM THE GENERATIVE PERSPECTIVE

In this section, we first redefine the objective of signaling games as (beta-)VAE's (Kingma & Welling, 2014; Higgins et al., 2017), i.e., the evidence lower bound (ELBO) with a KL weighting hyperparameter $\beta$. Next, we show several reasons supporting our ELBO-based formulation.

Though somewhat abrupt, let us think of a receiver agent $R_{\theta}$ as a generative model, a joint distribution over objects $X$ and messages $M$:

$$R_{\theta}^{\mathrm{joint}}(X, M) := R_{\theta}(X \mid M) P_{\theta}^{\mathrm{prior}}(M). \quad (6)$$

Intuitively, the receiver consists not only of the conventional reconstruction term $R_{\theta}(X|M)$ but also of a "language model" $P_{\theta}^{\mathrm{prior}}(M)$. The optimal sender would be the true posterior $S_{\theta}^*(M|X) = R_{\theta}(X|M) P_{\theta}^{\mathrm{prior}}(M)/P_{\mathrm{obj}}(X)$ by Bayes' theorem. Let us assume, however, that the sender can only approximate it via a variational posterior $S_{\phi}(M|X)$. The sender cannot access the true posterior because it is intractable, and the sender cannot directly peer into the receiver's "brain" $\theta$. Now one notices the similarity between the signaling game and (beta-)VAE.[3] Their similarity is illustrated in Figure 1 and Table 1. beta-VAE consists of an encoder, decoder, and prior latent distribution.

---

[2] In EC, *compositionality* is frequently studied (Kottur et al., 2017; Chaabouni et al., 2020; Li & Bowling, 2019; Andreas, 2019; Ren et al., 2020). The standard compositionality measure is *Topographic Similarity* (TopSim, Brighton & Kirby, 2006; Lazaridou et al., 2018). Let $d_{\mathcal{X}}$ be a distance function in an object space and $d_{\mathcal{M}}$ in a message space. TopSim is defined as the Spearman correlation between $d_{\mathcal{X}}(x^{(1)}, x^{(2)})$ and $d_{\mathcal{M}}(m^{(1)}, m^{(2)})$ for all combinations $x^{(1)}, x^{(2)} \in \mathcal{X}$ (without repetition). Here, each $m^{(i)}$ is the greedy sample from $S_{\phi}(M|x^{(i)})$. $d_{\mathcal{X}}$ and $d_{\mathcal{M}}$ are usually defined as Hamming and Levenshtein distance, respectively.

[3] The latent variable of the standard (beta-)VAE is a real vector in Euclidean space, whereas the latent variable of the signaling game is a discrete sequence.

Here, we regard $M$ as latent, $S_{\phi}(M|X)$ as encoder, $R_{\theta}(X|M)$ as receiver, and $P_{\theta}^{\text{prior}}(M)$ as prior distribution. Our objective is now defined as follows:

$$\mathcal{J}_{\text{ours}}(\phi, \theta; \beta) := \mathbb{E}_{x \sim P_{\text{obj}}(\cdot)}[\mathbb{E}_{m \sim S_{\phi}(\cdot|x)}[\log R_{\theta}(x \mid m)] - \beta D_{\text{KL}}(S_{\phi}(M \mid x) \| P_{\theta}^{\text{prior}}(M))], \quad (7)$$

where $D_{\text{KL}}$ denotes the KL divergence and $\beta \geq 0$ is a hyperparameter that weights the KL term. Although Eq. 7 equals to the precise ELBO only if $\beta = 1$, $\beta$ is often set $< 1$ initially and (optionally) annealed to 1 to prevent posterior collapse (Bowman et al., 2016; Alemi et al., 2018; Fu et al., 2019; Klushyn et al., 2019). In what follows, we discuss reasons supporting our redefinition.

---

**Recipe for Supporting Our ELBO-based Formulation**

1. **The conventional objective actually resembles ELBO in some respects.**

    Section 3.1: The conventional objective has an **implicit prior distribution**.

    Section 3.2: The conventional objective is **similar to beta-VAE's**.

2. **The choice of prior distributions can affect the properties of emergent languages.**

    Section 3.3: Be aware that **unnatural** prior distributions cause **inefficient** messages.

    Section 3.4: Use a **learnable** prior distribution instead, as a richer "**language model**."

    Section 4.2: The learnable prior distribution gives rise to **more meaningful segments**.

3. **ELBO can be justified from the computational psycholinguistic viewpoint.**

    Section 3.5: ELBO models the **trade-off** between **informativeness** and **surprisal**.

---

### 3.1 CONVENTIONAL OBJECTIVE HAS AN IMPLICIT PRIOR DISTRIBUTION

We show that the conventional signaling game has implicit prior distributions $P_{\text{unif}}^{\text{prior}}(M)$ or $P_{\alpha}^{\text{prior}}(M)$. First, let $P_{\text{unif}}^{\text{prior}}(M)$ be a uniform message distribution and $P_{\theta,\text{unif}}^{\text{joint}}(X, M)$ be as follows:

$$P_{\text{unif}}^{\text{prior}}(m) := \frac{1}{|\mathcal{M}|}, \quad P_{\theta,\text{unif}}^{\text{joint}}(x, m) := R_{\theta}(x \mid m) P_{\text{unif}}^{\text{prior}}(m). \quad (8)$$

Then, the following holds (see Appendix B.1 for its proof):

$$\nabla_{\phi, \theta} \mathcal{J}_{\text{conv}}(\phi, \theta) = \nabla_{\phi, \theta} \mathbb{E}_{x \sim P_{\text{obj}}(\cdot), m \sim S_{\phi}(\cdot|x)} \left[ \log P_{\theta,\text{unif}}^{\text{joint}}(x, m) \right], \quad (9)$$

i.e., maximizing $\mathcal{J}_{\text{conv}}(\phi, \theta)$ via the gradient method is the same as maximizing the expectation of $\log P_{\theta,\text{unif}}^{\text{joint}}(x, m)$. Next, the length-penalizing objective $\mathcal{J}_{\text{conv}}(\phi, \theta; \alpha)$ also has a prior distribution. Let $P_{\alpha}^{\text{prior}}(M)$ and $P_{\theta,\alpha}^{\text{joint}}(X, M)$ be as follows:

$$P_{\alpha}^{\text{prior}}(m) \propto \exp(-\alpha|m|), \quad P_{\theta,\alpha}^{\text{joint}}(x, m) := R_{\theta}(x \mid m) P_{\alpha}^{\text{prior}}(m). \quad (10)$$

Then, the following holds (see Appendix B.1 for its proof):

$$\nabla_{\phi, \theta} \mathcal{J}_{\text{conv}}(\phi, \theta; \alpha) = \nabla_{\phi, \theta} \mathbb{E}_{x \sim P_{\text{obj}}(\cdot), m \sim S_{\phi}(\cdot|x)}[\log P_{\theta,\alpha}^{\text{joint}}(x, m)], \quad (11)$$

i.e., maximizing $\mathcal{J}_{\text{conv}}(\phi, \theta; \alpha)$ is the same as maximizing the expectation of $\log P_{\theta,\alpha}^{\text{joint}}(x, m)$. Note that Eq. 8 is a special case of Eq. 10; if $\alpha = 0$, $P_{\text{unif}}^{\text{prior}}(m) = P_{\alpha}^{\text{prior}}(m)$ and $P_{\theta,\text{unif}}^{\text{joint}}(x, m) = P_{\theta,\alpha}^{\text{joint}}(x, m)$.

### 3.2 CONVENTIONAL OBJECTIVE IS SIMILAR TO (BETA-)VAE

The conventional objective may be similar to variational inference on a generative model, because it turned out to have implicit prior distributions. Indeed, it is similar to (beta-)VAE. As $\mathcal{J}_{\text{conv}}(\phi, \theta; \alpha)$ has the implicit prior $P_{\alpha}^{\text{prior}}(M)$, one notices the similarity between the signaling game and VAE, regarding $M$ as latent, $S_{\phi}(M|X)$ as encoder, $R_{\theta}(X|M)$ as decoder. Here, the corresponding ELBO $\mathcal{J}_{\text{elbo}}(\phi, \theta; \alpha)$ can be written as follows:

$$\mathcal{J}_{\text{elbo}}(\phi, \theta; \alpha) := \mathbb{E}_{x \sim P_{\text{obj}}(\cdot)}[\mathbb{E}_{m \sim S_{\phi}(\cdot|x)}[\log R_{\theta}(x \mid m)] - D_{\text{KL}}(S_{\phi}(M|x) \| P_{\alpha}^{\text{prior}}(M))]. \quad (12)$$

Then, the following holds (see Appendix B.2 for its proof):

$$\nabla_{\phi,\theta}\mathcal{J}_{\text{elbo}}(\phi,\theta;\alpha) = \nabla_{\phi,\theta}\mathcal{J}_{\text{conv}}(\phi,\theta;\alpha) + \underbrace{\nabla_{\phi,\theta}\mathbb{E}_{x\sim P_{\text{obj}}(\cdot)}[\mathcal{H}(S_{\phi}(M\mid x))]}_{\text{resembles }\nabla_{\phi,\theta}^{\text{entreg}}\text{ (Eq. 3)}}, \qquad (13)$$

i.e., maximizing $\mathcal{J}_{\text{elbo}}(\phi,\theta;\alpha)$ is the same as maximizing $\mathcal{J}_{\text{conv}}(\phi,\theta;\alpha)$ plus an *entropy maximizer*. Recall that previous work typically adopts an *entropy regularizer* (Eq. 3). Though they are not mathematically equal, they are similar enough in that they keep the sender's entropy high to encourage exploration (Levine, 2018). In this sense, the conventional signaling game is similar to VAE. Or rather, it is similar to beta-VAE because $\lambda^{\text{entreg}}$ adjusts the entropy regularizer in practice, which roughly corresponds to that $\beta$ adjusts the KL term weight.

### 3.3 Implicit Prior Distribution and Inefficiency of Emergent Language

We have thus far seen that the conventional objective has an implicit prior distribution and is similar to beta-VAE. Now we need to elaborate on *why* one should be aware of them. To this end, in this subsection, we suggest that the inefficiency of emergent languages is due to *unnatural* prior distributions. Chaabouni et al. (2019) reported that the objective $\mathcal{J}_{\text{conv}}(\phi,\theta)$ did not result in emergent languages that follow ZLA, while the length-penalizing one $\mathcal{J}_{\text{conv}}(\phi,\theta;\alpha)$ did. Recall that $\mathcal{J}_{\text{conv}}(\phi,\theta)$ has the implicit prior distribution $P_{\text{unif}}^{\text{prior}}(M)$ and $\mathcal{J}_{\text{conv}}(\phi,\theta;\alpha)$ has $P_{\alpha}^{\text{prior}}(M)$. Then, we define the corresponding distributions over message lengths $K$:

$$\begin{aligned}
P_{\text{unif}}^{\text{prior}}(K=k) &:= \sum_{m\in\mathcal{M}}\mathbb{1}_{|m|=k}P_{\text{unif}}^{\text{prior}}(m) \propto \exp(\gamma k),\\
P_{\alpha}^{\text{prior}}(K=k) &:= \sum_{m\in\mathcal{M}}\mathbb{1}_{|m|=k}P_{\alpha}^{\text{prior}}(m) \propto \exp((\gamma-\alpha)k),
\end{aligned} \qquad (14)$$

where $\gamma = \log(|\mathcal{A}|-1) > 0$ and $\mathbb{1}_{(\cdot)}$ is the indicator function. $P_{\text{unif}}^{\text{prior}}(k)$ (resp. $P_{\alpha}^{\text{prior}}(k)$) defines the probability that the length $K$ of a message sampled from $P_{\text{unif}}^{\text{prior}}(M)$ (resp. $P_{\alpha}^{\text{prior}}(M)$) is $k$. $P_{\text{unif}}^{\text{prior}}(k)$ grows up w.r.t $k$, i.e., longer messages are more common, and even the longest messages are the most likely. It is clearly unnatural as a prior distribution. Regularized by such a prior distribution, emergent languages would obviously become unnaturally long. In contrast, when $\alpha > \gamma$, $P_{\alpha}^{\text{prior}}(k)$ decays w.r.t $k$, i.e., shorter messages are more common. Thus, unnatural inefficiency is arguably due to the implicit, inappropriate prior distribution $P_{\text{unif}}^{\text{prior}}(m)$, while $P_{\alpha}^{\text{prior}}(m)$ may mitigate the problem. This conclusion contrasts with the previous work that has ascribed the issue to inductive biases, as mentioned in Section 2.2. Note that it is an *additional interpretation* rather than a rebuttal. It is likely to be a complicated issue involving both the prior distribution and inductive bias.

### 3.4 Introduce a Learnable Parametrized Prior Distribution

In the previous subsection, we saw that the choice of prior distributions can influence the statistical properties of emergent languages. However, the signaling-game counterpart of the VAE's prior distribution has not been clear yet. In other words, as a sub-domain of linguistics, it is important to take a clear stance on which part of real-world communication the prior distribution models. This paper considers it as a receiver's neural *language model* (LM) parametrized by $\theta$:

$$P_{\theta}^{\text{prior}}(m) = \prod_{t=1}^{|m|} P_{\theta}^{\text{prior}}(m_t \mid m_{1:t-1}). \qquad (15)$$

It can be seen as an LM because it approximates the average behavior of a sender, i.e., the KL term in Eq. 7, $\mathbb{E}_{P_{\text{obj}}(x)}[D_{\text{KL}}(S_{\phi}(M|x)\|P_{\theta}^{\text{prior}}(M))] = -\mathbb{E}_{P_{\text{obj}}(x)}[\mathcal{H}(S_{\phi}(M|x))] - \mathbb{E}_{S_{\phi}(m)}[\log P_{\theta}^{\text{prior}}(m)]$, is minimized w.r.t $\theta$ when $P_{\theta}^{\text{prior}}(m) = \mathbb{E}_{P_{\text{obj}}(x)}[S_{\phi}(m|x)]$. In contrast, $P_{\alpha}^{\text{prior}}(m)$ seems to be oversimplified as an LM; though it is more natural than $P_{\text{unif}}^{\text{prior}}(m)$, it is still unnatural as a simulation of language since it restricts the complexity of the sender's behavior. Our modification leads to more meaningful segments (see Section 4).

### 3.5 Relationship to Surprisal Theory

We formulated the game objective as ELBO and defined its prior distribution as a learnable LM. It can also be seen as a modeling of *surprisal theory* (Hale, 2001; Levy, 2008; Smith & Levy, 2013) in

computational psycholinguistics. The theory has recently been considered important for evaluating neural language models as cognitive models (Wilcox et al., 2020). In the theory, *surprisal* refers to the negative log-likelihood of sentences/messages $-\log P_{\boldsymbol{\theta}}^{\mathrm{prior}}(m) = -\sum_t \log P_{\boldsymbol{\theta}}^{\mathrm{prior}}(m_t|m_{1:t-1})$ and higher surprisals are considered to require greater cognitive processing costs. Intuitively, the cost for a reader/listener/receiver to predict the next token accumulates over time. In fact, our objective $\mathcal{J}_{\mathrm{ours}}$ models it naturally, as one notices from the following rewrite:

$$\mathcal{J}_{\mathrm{ours}}(\boldsymbol{\phi}, \boldsymbol{\theta}; \beta) = \mathbb{E}_{x \sim P_{\mathrm{obj}}(\cdot)}[\mathbb{E}_{m \sim S_{\boldsymbol{\phi}}(\cdot|x)}[\underbrace{\log R_{\boldsymbol{\theta}}(x \mid m)}_{\text{reconstruction}} + \underbrace{\beta \log P_{\boldsymbol{\theta}}^{\mathrm{prior}}(m)}_{\text{negative surprisal}}] + \underbrace{\beta \, \mathcal{H}(S_{\boldsymbol{\phi}}(M \mid x))}_{\text{entropy maximizer}}]. \quad (16)$$

The first term is the reconstruction, the second is the (negative) surprisal, and the third is the entropy maximizer. Reconstruction and surprisal can be in a trade-off relationship. Conveying more information results in messages with a higher cognitive cost, while messages with a lower cognitive cost may not convey as much information.[4]

## 4 EXPERIMENT

In this section, we validate the effects of the formulation presented in Section 3. Specifically, we demonstrate improvements in response to the problem raised by Ueda et al. (2023) that the conventional objective does not give rise to meaningful word segmentation in emergent languages. The experimental setup is described in Section 4.1 and the results are presented in Section 4.2.[5]

### 4.1 SETUP

**Object and Message Spaces:** We define an object space $\mathcal{X}$ as a set of att-val objects. Following Ueda et al. (2023), we set $(n_{\mathrm{att}}, n_{\mathrm{val}}) \in \{(2, 64), (3, 16), (4, 8), (6, 4), (12, 2)\}$, ensuring all the object space sizes $|\mathcal{X}|$ are $(n_{\mathrm{val}})^{n_{\mathrm{att}}} = 4096$.[6] To define a message space $\mathcal{M}$, we set $L_{\max} = 32$, the same as Ueda et al. (2023), and $|\mathcal{A}| = 9$, one bigger than Ueda et al. (2023) so that $\mathtt{eos} \in \mathcal{A}$.

**Agent Architectures and Optimization:** The agents $S_{\boldsymbol{\phi}}, R_{\boldsymbol{\theta}}$ are based on GRU (Cho et al., 2014), following Chaabouni et al. (2020); Ueda et al. (2023) and have symbol embedding layers. Layer-Norm (Ba et al., 2016) is applied to the hidden states and embeddings for faster convergence. The agents have linear layers (with biases) to predict the next symbol; Using the symbol prediction layer, the sender generates a message while the receiver computes $\log P_{\boldsymbol{\theta}}^{\mathrm{prior}}(m)$. The sender embeds an object $x$ to the initial hidden state, and the receiver outputs the object logits from the last hidden state. We apply Gaussian dropout (Srivastava et al., 2014) to regularize $\boldsymbol{\theta}$, because otherwise it can be unstable under competing pressures $\log R_{\boldsymbol{\theta}}(m|x)$ and $\log R_{\boldsymbol{\theta}}(m)$. We optimize ELBO in which latent variables are discrete sequences, similarly to Mnih & Gregor (2014). By the stochastic computation graph approach (Schulman et al., 2015), we obtain a surrogate objective that is the combination of the standard backpropagation w.r.t the receiver's parameter $\boldsymbol{\theta}$ and REINFORCE (Williams, 1992) w.r.t the sender's parameter $\boldsymbol{\phi}$. Consequently, we obtain a similar optimization strategy to the one adopted in Chaabouni et al. (2019); Ueda et al. (2023). $\beta$ is initially set $\ll 1$ to avoid posterior collapse and annealed to 1 via REWO (Klushyn et al., 2019). For more detail, see Appendix C.

**Evaluation:** We have three criteria C1, C2, and C3 to measure the meaningfulness of segments obtained by the boundary detection algorithm. The threshold parameter is set to 0.[7] We compare our objective with the following baselines: (BL1) the conventional objective $\mathcal{J}_{\mathrm{conv}}$ plus the entropy regularizer (where $\lambda^{\mathrm{entreg}} = 1$), (BL2) the ELBO-based objective that is almost the same as ours, except that its prior is $P_{\alpha}^{\mathrm{prior}}$ (where $\alpha = \log|\mathcal{A}|$).[8] We ran each 16 times with distinct random seeds.

---

[4]It may also be seen as the trade-off between expressivity/informativeness and compressivity/efficiency in the language evolution field (Kirby et al., 2015; Zaslavsky et al., 2022), though our ELBO-based formulation *per se* is orthogonal to models involving generation change, e.g., iterated learning (Kirby et al., 2014).

[5]We also investigated whether emergent languages show a ZLA-like tendency when using our objective. They indeed show such a tendency with moderate alphabet sizes. See Appendix E.

[6]We exclude $(1, 4096)$ because, in preliminary experiments, the learning convergence was much slower. It is not a big concern as it is an exceptional configuration for which TopSim cannot be defined.

[7]We also tried $0.25, 0.5$ for the threshold parameter (see Appendix D).

[8]When $\alpha = \log|\mathcal{A}|$, $\lim_{L_{\max} \to \infty} P_{\alpha}^{\mathrm{prior}}(M)$ coincides with a *uniform monkey typing* (see Appendix B.3).

Figure 2: Results for $n_{\text{bou}}$ (C1), $n_{\text{seg}}$ (C2), $\Delta_{\text{w,c}}$ (C3), C-TopSim (C3), and W-TopSim (C3) are shown in order from the left. The x-axis represents $(n_{\text{att}}, n_{\text{val}})$ while the y-axis represents the values of each metric. The shaded regions and error bars represent the standard error of mean. The threshold parameter is set to $0$. The blue plots represent the results for our ELBO-based objective $\mathcal{J}_{\text{ours}}$, the orange ones for (BL1) the conventional objective $\mathcal{J}_{\text{conv}}$ plus the entropy regularizer, and the grey ones for (BL2) the ELBO-based objective whose prior is $P_{\alpha}^{\text{prior}}$. The apparent inferior performance of $\Delta_{\text{w,c}}$ for $\mathcal{J}_{\text{ours}}$ compared to the baselines might be misleading. It is because $\mathcal{J}_{\text{ours}}$ greatly improves both C-TomSim and W-TopSim. The larger scale of their improvements could result in a seemingly worse $\Delta_{\text{w,c}}$, but this does not necessarily indicate poorer performance.

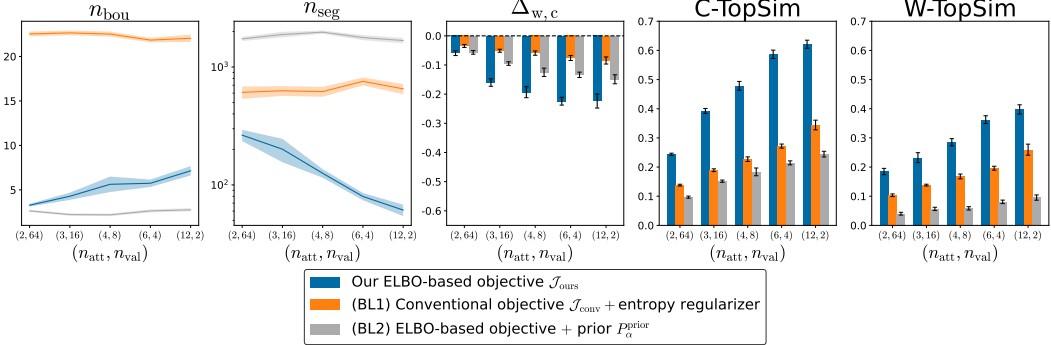

## 4.2 EXPERIMENTAL RESULT

We show the results in Figure 2. The results for $n_{\text{bou}}$ (C1), $n_{\text{seg}}$ (C2), $\Delta_{\text{w,c}}$ (C3), C-TopSim (C3), and W-TopSim (C3) are shown in order from the left. As will be explained below, it can be observed that the meaningfulness of segments in emergent language improves when using our objective $\mathcal{J}_{\text{ours}}$.

**Criteria C1 and C2 are satisfied with our objective $\mathcal{J}_{\text{ours}}$:** First, see the result for $n_{\text{bou}}$ in Figure 2. It is evident that the line monotonically increases only for $\mathcal{J}_{\text{ours}}$. It means that, as $n_{\text{att}}$ increases, $n_{\text{bou}}$ also increases only when using $\mathcal{J}_{\text{ours}}$, confirming that C1 is satisfied. Next, see the result for $n_{\text{seg}}$ in Figure 2. Again, it is evident that the line monotonically decreases only for $\mathcal{J}_{\text{ours}}$. That is, as $n_{\text{val}}$ increases, $n_{\text{seg}}$ also increases only when using $\mathcal{J}_{\text{ours}}$, confirming that C2 is satisfied.

**Criterion C3 is not met but TopSim improved:** Refer to the results of $\Delta_{\text{w,c}}$ in Figure 2. $\Delta_{\text{w,c}} < 0$ regardless of the objective used. Even when using $\mathcal{J}_{\text{ours}}$, C3 is not satisfied. However, observe the results for C-TopSim and W-TopSim in Figure 2. By using $\mathcal{J}_{\text{ours}}$, both C-TopSim and W-TopSim have improved. Notably, the improvement in W-TopSim indicates that the meaningfulness of segments has improved. $\Delta_{\text{w,c}}$ does not become positive since C-TopSim has increased even more. We speculate that one reason for $\Delta_{\text{w,c}} < 0$ might be due to the "spelling variation" of segments.

## 5 DISCUSSION

In Section 3, we proposed formulating the signaling game objective as ELBO and discussed supporting scenarios. In particular, we suggested that prior distributions can influence whether emergent languages follow ZLA. In Section 4, we demonstrated the improvement of segments' meaningfulness regarding HAS with a learnable prior distribution (neural language model).

**Why Did Segments Become More Meaningful?** Though it is not straightforward to give a clear mathematical basis, we currently hypothesize that the improvement of segments' meaningfulness is due to the competing pressure of the reconstruction and the surprisal. On the one hand, due to the reconstruction term, a receiver has to be "surprised," i.e., to receive symbols $m_t$ with relatively high $-\log P_{\boldsymbol{\theta}}^{\text{prior}}(m_t | m_{1:t-1})$. Objects contain $n_{\text{att}}$ statistically independent components, i.e., attributes. The receiver must be surprised several times $\propto n_{\text{att}}$ to reconstruct the object from a sequential message. On the other hand, due to the surprisal term, the receiver does not want to be "surprised," i.e., it prefers relatively low $-\log P_{\boldsymbol{\theta}}^{\text{prior}}(m_t | m_{1:t-1})$. In natural language, branching entropy (Eq. 5) decreases on average, but there are some spikes indicating boundaries. In other words, the next character should often be predictable (less surprising), while boundaries appear at points where

predictions are hard. The conventional objective $\mathcal{J}_{\text{conv}}$ does not provide any incentives to make characters "often predictable," whereas the surprisal term in ELBO provides.[9]

**Limitation and Future Work:** A populated signaling game (Chaabouni et al., 2022; Rita et al., 2022a; Michel et al., 2023) that involves multiple senders and receivers is important from a sociolinguistic perspective (Raviv et al., 2019). Defining a reasonable ELBO for such a setting is our future work. Perhaps it would be similar to multi-view VAE (Suzuki & Matsuo, 2022). Another remaining task is to extend our generative perspective to discrimination games for which the objective is not reconstructive but contrastive, such as InfoNCE (van den Oord et al., 2018). Furthermore, although we adopted GRU as a prior message distribution, various neural language models have been proposed as cognitive models in computational (psycho)linguistics (Dyer et al., 2016; Shen et al., 2019; Stanojevic et al., 2021; Kuribayashi et al., 2022, *inter alia*). It should be a key direction to investigate whether such cognitively motivated models give rise to richer structures such as syntax.

## 6 RELATED WORK

**EC as Representation Learning:** Several studies regard EC as representation learning (Andreas, 2019; Chaabouni et al., 2020; Xu et al., 2022). For instance, compositionality and disentanglement are often seen as synonymous. TopSim is a representation similarity analysis (RSA, Kriegeskorte et al., 2008), as van der Wal et al. (2020) pointed out. Resnick et al. (2020) formulated the objective as ELBO. However, they defined messages as fixed-length binaries and agents as non-autoregressive models, which is not applicable to our variable-length setting. Moreover, they do not discuss the prior distribution choice; they seem to have adopted Bernoulli purely for computational convenience. In contrast, we indicated that their choice influences the structure of emergent languages.[10]

**VIB:** Tucker et al. (2022) defined a communication game called *VQ-VIB*, based on Variational Information Bottleneck (VIB, Alemi et al., 2017) which is known as a generalization of beta-VAE. Also, Chaabouni et al. (2021) formalized a *color naming game* with a similar motivation. Note that messages are defined as single symbols in their settings. In contrast, messages are of variable length in our setting so that we can discuss the structure of emergent languages such as ZLA and HAS.

**MH Naming Game:** Taniguchi et al. (2022), Inukai et al. (2023), and Hoang et al. (2024) defined a (recursive) *MH naming game* in which communication was formulated as representation learning on a generative model and emerged through MCMC instead of variational inference.

**Natural Language as Prior:** Havrylov & Titov (2017) formalized a referential game with a pretrained natural language model as a prior distribution. Their qualitative evaluation showed that languages became more compositional. Lazaridou et al. (2020) adopted a similar approach.

**Dealing with Discreteness:** Though we adopted the REINFORCE-like optimization following the previous EC work (Chaabouni et al., 2019; Ueda et al., 2023), there are various approaches dealing with discrete variables (Rolfe, 2017; Vahdat et al., 2018a;b; Jang et al., 2017; Maddison et al., 2017).

## 7 CONCLUSION

In this paper, EC was reinterpreted as representation learning on a generative model. We regarded Lewis's signaling game as beta-VAE and formulated its objective as ELBO. Our main contributions are (1) to show the similarities between the game and (beta-)VAE, (2) to indicate the impact of prior message distribution on the statistical properties of emergent languages, and (3) to present a computational psycholinguistic interpretation of emergent communication. Specifically, we addressed the issues of Zipf's law of abbreviation and Harris's articulation scheme, which had not been reproduced well with the conventional objective. Another important point is that we revisited the variable-length message setting, which previous work often avoided. The remaining tasks for future work are formulating a populated or discrimination game (while keeping the generative viewpoint) and introducing cognitively motivated language models as prior distributions.

---

[9]Our current hypothesis does not fully explain the reason why the branching entropy does not converge to a flat one instead of a jittering one. At the moment, we conjecture that, as a matter of probability, it is simply difficult to converge to a flat one among infinitely many possible bit patterns (almost all of them would jitter).

[10]In Appendix F, Table 2 supplementally shows the relationship between previous EC work and this paper.

ACKNOWLEDGMENTS

This work was supported by JSPS KAKENHI Grant Numbers JP21H04904, JP23H04835, and JP23KJ0768. We would like to express our gratitude to the anonymous reviewers for their lively and constructive discussions.

REPRODUCIBILITY STATEMENT

The reproducibility of the experiments in this paper is ensured in the following ways:

- The experimental setup is described in Section 4.1 and Appendix C.
- The source code has been released upon acceptance (See Appendix C).

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

## A    HAS-BASED BOUNDARY DETECTION ALGORITHM

---

**Algorithm 1** Boundary Detection Algorithm

---
1:   $i \leftarrow 0;\ w \leftarrow 1;\ \mathcal{B} \leftarrow \{\}$
2: **while** $i < n$ **do**
3:      Compute $\mathrm{BE}(s_{i:i+w-1})$
4:      **if** $\mathrm{BE}(s_{i:i+w-1}) - \mathrm{BE}(s_{i:i+w-2}) >$ threshold $\&\ w > 1$ **then**
5:          $\mathcal{B} \leftarrow \mathcal{B} \cup \{i + w\}$
6:      **end if**
7:      **if** $i + w < n - 1$ **then**
8:          $w \leftarrow w + 1$
9:      **else**
10:         $i \leftarrow i + 1;\ w \leftarrow 1$
11:      **end if**
12: **end while**

---

We show the HAS-based boundary detection algorithm (Tanaka-Ishii, 2005; Tanaka-Ishii & Ishii, 2007) in Algorithm 1. Note that this pseudo code is a simplified version by Ueda et al. (2023). $\mathcal{B}$ is a set of boundary positions. threshold is a hyperparameter.

# B PROOFS ON THE EXISTENCE OF IMPLICIT PRIORS

## B.1 PROOF OF EQ. 9 AND EQ. 11

*Remark* 1. Since $P_\alpha^{\text{prior}}(m \mid \alpha) \propto \exp(-\alpha|m|)$ by definition,

$$P_\alpha^{\text{prior}}(m \mid \alpha) = \frac{1}{Z_\alpha} \exp(-\alpha|m|),$$

where $Z_\alpha$ is a normalizer to ensure that $P_\alpha^{\text{prior}}(M \mid \alpha)$ is a probability distribution. $Z_\alpha$ can be written as follows:

$$
\begin{aligned}
Z_\alpha &= \sum_{m \in \mathcal{M}} \exp(-\alpha|m|) \\
&= \exp(-\alpha) \sum_{l=1}^{L_{\max}} \{(|\mathcal{A}| - 1) \exp(-\alpha)\}^{l-1} \quad \text{(There are } (|\mathcal{A}| - 1)^{l-1} \text{ messages of length } l.) \\
&= \exp(-\alpha) \frac{1 - \{(|\mathcal{A}| - 1) \exp(-\alpha)\}^{L_{\max}}}{1 - (|\mathcal{A}| - 1) \exp(-\alpha)} \\
&= \frac{1 - \{(|\mathcal{A}| - 1) \exp(-\alpha)\}^{L_{\max}}}{\exp(\alpha) - |\mathcal{A}| + 1}.
\end{aligned}
$$

**Lemma 1.** *The following equation holds:*

$$
\begin{aligned}
&\nabla_{\phi,\theta} \, \mathbb{E}_{x \sim P_{obj}(\cdot), S_\phi(\cdot|x)}[f_\theta(x, m)] \\
&= \mathbb{E}_{x \sim P_{obj}(\cdot), S_\phi(\cdot|x)}[\nabla_{\phi,\theta} f_\theta(x, m) + f_\theta(x, m) \nabla_{\phi,\theta} \log S_\phi(m \mid x)],
\end{aligned}
$$

*where $f_\theta : \mathcal{X} \times \mathcal{M} \to \mathbb{R}$ is an any function differentiable w.r.t $\theta$.*

**Proof of Lemma 1.**
Let $\text{supp}(P)$ be the support of a given probability measure $P$. For the notational convenience, we define $\mathcal{M}'_x := \text{supp}(S_\phi(M|x))$. Then,

$$
\begin{aligned}
&\nabla_{\phi,\theta} \, \mathbb{E}_{x \sim P_{\text{obj}}(\cdot), m \sim S_\phi(\cdot|x)}[f_\theta(x, m)] \\
&= \nabla_{\phi,\theta} \sum_{x \in \mathcal{X}} \sum_{m \in \mathcal{M}'_x} P_{\text{obj}}(x) S_\phi(m \mid x) f_\theta(x, m) \\
&= \sum_{x \in \mathcal{X}} \sum_{m \in \mathcal{M}'_x} P_{\text{obj}}(x) \left( S_\phi(m \mid x) \nabla_{\phi,\theta} f_\theta(x, m) + \nabla_{\phi,\theta} S_\phi(m \mid x) f_\theta(x, m) \right) \\
&= \sum_{x \in \mathcal{X}} \sum_{m \in \mathcal{M}'_x} P_{\text{obj}}(x) \left( S_\phi(m \mid x) \nabla_{\phi,\theta} f_\theta(x, m) + S_\phi(m \mid x) \frac{\nabla_{\phi,\theta} S_\phi(m \mid x)}{S_\phi(m \mid x)} f_\theta(x, m) \right) \\
&= \sum_{x \in \mathcal{X}} \sum_{m \in \mathcal{M}'_x} P_{\text{obj}}(x) \left( S_\phi(m \mid x) \nabla_{\phi,\theta} f_\theta(x, m) + S_\phi(m \mid x)(\nabla_{\phi,\theta} \log S_\phi(m \mid x)) f_\theta(x, m) \right) \\
&= \sum_{x \in \mathcal{X}} \sum_{m \in \mathcal{M}'_x} P_{\text{obj}}(x) S_\phi(m \mid x) \left( \nabla_{\phi,\theta} f_\theta(x, m) + f_\theta(x, m) \nabla_{\phi,\theta} \log S_\phi(m \mid x) \right) \\
&= \mathbb{E}_{x \sim P_{\text{obj}}(\cdot), m \sim S_\phi(\cdot|x)}[\nabla_{\phi,\theta} f_\theta(x, m) + f_\theta(x, m) \nabla_{\phi,\theta} \log S_\phi(m \mid x)].
\end{aligned}
$$

$\square$

*Remark* 2. As Chaabouni et al. (2019) pointed out, Lemma 1 can be seen as a special case of the stochastic computation graph approach by Schulman et al. (2015).

**Lemma 2.** *The following equation holds:*

$$\mathbb{E}_{x \sim P_{obj}(\cdot), S_\phi(\cdot|x)}[g(x) \nabla_{\phi,\theta} \log S_\phi(m \mid x)] = 0,$$

*where $g : \mathcal{X} \to \mathbb{R}$ is an any function. The function $g$ is called baseline in the literature of reinforcement learning.*

**Proof of Lemma 2.**
Let $\text{supp}(P)$ be the support of a given probability measure $P$. For the notational convenience, we

define $\mathcal{M}'_x := \text{supp}(S_\phi(M|x))$. Then,

$$\mathbb{E}_{x\sim P_{\text{obj}}(\cdot),mS_\phi(\cdot|x)}[g(x)\nabla_{\phi,\theta}\log S_\phi(m\mid x)]$$

$$= \sum_{x\in\mathcal{X}} P_{\text{obj}}(x)g(x)\sum_{m\in\mathcal{M}'_x} S_\phi(m|x)\nabla_{\phi,\theta}\log S_\phi(m\mid x)$$

$$= \sum_{x\in\mathcal{X}} P_{\text{obj}}(x)g(x)\sum_{m\in\mathcal{M}'_x} S_\phi(m|x)\frac{\nabla_{\phi,\theta}S_\phi(m\mid x)}{S_\phi(m\mid x)}$$

$$= \sum_{x\in\mathcal{X}} P_{\text{obj}}(x)g(x)\sum_{m\in\mathcal{M}'_x} \nabla_{\phi,\theta}S_\phi(m\mid x)$$

$$= \sum_{x\in\mathcal{X}} P_{\text{obj}}(x)g(x)\nabla_{\phi,\theta}\left(\sum_{m\in\mathcal{M}'_x} S_\phi(m\mid x)\right)$$

$$= \sum_{x\in\mathcal{X}} P_{\text{obj}}(x)g(x)\nabla_{\phi,\theta}1 = 0.$$

$\square$

**Proof of Eq. 9 and Eq. 11.**

As $P_{\text{unif}}^{\text{prior}}(M)$ is a special case of $P_\alpha^{\text{prior}}(M\mid\alpha)$, it is sufficient to prove Eq. 11.

Recall that what we have to prove, i.e., Eq. 11, is as follows:

$$\nabla_{\phi,\theta}\mathcal{J}_{\text{conv}}(\phi,\theta;\alpha) = \nabla_{\phi,\theta}\mathbb{E}_{x\sim P_{\text{obj}}(\cdot),m\sim S_\phi(\cdot|x)}[\log P_{\theta,\alpha}^{\text{joint}}(x,m)],$$

where

$$P_\alpha^{\text{prior}}(m\mid\alpha) = \frac{1}{Z_\alpha}\exp(-\alpha|m|),$$

$$P_{\theta,\alpha}^{\text{joint}}(x,m) = R_\theta(x\mid m)P_\alpha^{\text{prior}}(m).$$

Here, let us define $T$ (for convenience) as:

$$T := \mathbb{E}_{x\sim P_{\text{obj}}(\cdot),m\sim S_\phi(\cdot|x)}[\nabla_{\phi,\theta}\log R_\theta(x|m) + \{\log R_\theta(x|m) - \alpha|m|\}\nabla_{\phi,\theta}S_\phi(m\mid x)].$$

In what follows, we first prove the left-hand side of Eq. 11 is equal to $T$, i.e., $\nabla_{\phi,\theta}\mathcal{J}_{\text{conv}}(\phi,\theta;\alpha) = T$. We next prove that the right-hand side of Eq. 11 is equal to $T$, i.e., $\nabla_{\phi,\theta}\mathbb{E}_{x\sim P_{\text{obj}}(\cdot),m\sim S_\phi(\cdot|x)}[\log P_{\theta,\alpha}^{\text{joint}}(x,m)] = T$. We finally conclude that $\nabla_{\phi,\theta}\mathcal{J}_{\text{conv}}(\phi,\theta;\alpha) = T = \nabla_{\phi,\theta}\mathbb{E}_{x\sim P_{\text{obj}}(\cdot),m\sim S_\phi(\cdot|x)}[\log P_{\theta,\alpha}^{\text{joint}}(x,m)]$, and thus Eq. 11 holds.

On the one hand, the left-hand side of Eq. 11 can be transformed as follows:

$$\nabla_{\phi,\theta}\mathcal{J}_{\text{conv}}(\phi,\theta;\alpha)$$

$$= \nabla_{\phi,\theta}\mathbb{E}_{x\sim P_{\text{obj}}(\cdot),m\sim S_\phi(\cdot|x)}[\log R_\theta(x|m) - \alpha|m|]$$

$$\text{(By the definition of } \mathcal{J}_{\text{conv}}(\phi,\theta;\alpha))$$

$$= \mathbb{E}_{x\sim P_{\text{obj}}(\cdot),m\sim S_\phi(\cdot|x)}[\nabla_{\phi,\theta}\log R_\theta(x|m) - \alpha\nabla_{\phi,\theta}|m| + \{\log R_\theta(x|m) - \alpha|m|\}\nabla_{\phi,\theta}S_\phi(m|x)]$$

$$\text{(From Lemma 1, defining } f_\phi(x,m) := \log R_\phi(x\mid m) - \alpha|m|)$$

$$= \mathbb{E}_{x\sim P_{\text{obj}}(\cdot),m\sim S_\phi(\cdot|x)}[\nabla_{\phi,\theta}\log R_\theta(x|m) + \{\log R_\theta(x|m) - \alpha|m|\}\nabla_{\phi,\theta}S_\phi(m\mid x)]$$

$$\text{(because } \nabla_{\phi,\theta}|m| = 0)$$

$$= T.$$

On the other hand, the right-hand side of Eq. 11 can be transformed as follows:

$$\nabla_{\phi,\theta}\mathbb{E}_{x\sim P_{\text{obj}}(\cdot),m\sim S_\phi(\cdot|x)}[\log P_{\theta,\alpha}^{\text{joint}}(x,m\mid\alpha)]$$

$$= \nabla_{\phi,\theta}\mathbb{E}_{x\sim P_{\text{obj}}(\cdot),m\sim S_\phi(\cdot|x)}[\log R_\theta(x\mid m) + \log P_\alpha^{\text{prior}}(m\mid\alpha)]$$

$$\text{(By the definition of } \mathcal{J}_{\text{conv}}(\phi,\theta;\alpha))$$

$$= \mathbb{E}_{x\sim P_{\text{obj}}(\cdot),m\sim S_\phi(\cdot|x)}[$$
$$\nabla_{\phi,\theta}\log R_\theta(x|m) + \nabla_{\phi,\theta}\log P_\alpha^{\text{prior}}(m|\alpha) + \{\log R_\theta(x|m) + \log P_\alpha^{\text{prior}}(m|\alpha)\}\nabla_{\phi,\theta}\log S_\phi(m|x)]$$

$$(\text{From Lemma 1, defining } f_{\boldsymbol{\theta}}(x, m) := \log R_{\boldsymbol{\theta}}(m \mid x) + \log P_\alpha^{\text{prior}}(m \mid \alpha))$$

$$= \mathbb{E}_{x \sim P_{\text{obj}}(\cdot), m \sim S_{\boldsymbol{\phi}}(\cdot \mid x)} \big[$$
$$\nabla_{\boldsymbol{\phi}, \boldsymbol{\theta}} \log R_{\boldsymbol{\theta}}(x \mid m) + \{\log R_{\boldsymbol{\theta}}(x \mid m) - \alpha |m| - \log Z_\alpha\} \nabla_{\boldsymbol{\phi}, \boldsymbol{\theta}} \log S_{\boldsymbol{\phi}}(m \mid x)\big]$$

$$(\text{Because } \nabla_{\boldsymbol{\phi}, \boldsymbol{\theta}} \log P_\alpha^{\text{prior}}(m \mid \alpha) = 0 \text{ and } \log P_\alpha^{\text{prior}}(m \mid \alpha) = -\alpha |m| - \log Z_\alpha)$$

$$= \mathbb{E}_{x \sim P_{\text{obj}}(\cdot), m \sim S_{\boldsymbol{\phi}}(\cdot \mid x)} [\nabla_{\boldsymbol{\phi}, \boldsymbol{\theta}} \log R_{\boldsymbol{\theta}}(x \mid m) + \{\log R_{\boldsymbol{\theta}}(x \mid m) - \alpha |m|\} \nabla_{\boldsymbol{\phi}, \boldsymbol{\theta}} \log S_{\boldsymbol{\phi}}(m \mid x)]$$

$$(\text{From Lemma 2, defining } g(x) := \log Z_\alpha)$$

$$= T.$$

Therefore, Eq. 11 holds. For Eq. 9, consider the special case, i.e., $\alpha = 0$. □

## B.2 PROOF OF EQ. 13

Recall that $\mathcal{J}_{\text{elbo}}(\boldsymbol{\phi}, \boldsymbol{\theta}; \alpha)$ is defined as:

$$\mathcal{J}_{\text{elbo}}(\boldsymbol{\phi}, \boldsymbol{\theta}; \alpha) := \mathbb{E}_{x \sim P_{\text{obj}}(\cdot)}[\mathbb{E}_{m \sim S_{\boldsymbol{\phi}}(\cdot \mid x)}[\log R_{\boldsymbol{\theta}}(x \mid m)] - D_{\text{KL}}(S_{\boldsymbol{\phi}}(M | x) || P_\alpha^{\text{prior}}(M))].$$

Also, recall that what we have to prove, i.e., Eq. 13, is as follows:

$$\nabla_{\boldsymbol{\phi}, \boldsymbol{\theta}} \mathcal{J}_{\text{elbo}}(\boldsymbol{\phi}, \boldsymbol{\theta}; \alpha) = \nabla_{\boldsymbol{\phi}, \boldsymbol{\theta}} \mathcal{J}_{\text{conv}}(\boldsymbol{\phi}, \boldsymbol{\theta}; \alpha) + \nabla_{\boldsymbol{\phi}, \boldsymbol{\theta}} \mathbb{E}_{x \sim P_{\text{obj}}(\cdot)}[\mathcal{H}(S_{\boldsymbol{\phi}}(M \mid x))].$$

**Proof of Eq. 13.**
First, $\mathcal{J}_{\text{elbo}}(\boldsymbol{\phi}, \boldsymbol{\theta}; \alpha)$ can be transformed as follows:

$$\mathcal{J}_{\text{elbo}}(\boldsymbol{\phi}, \boldsymbol{\theta}; \alpha)$$
$$= \mathbb{E}_{x \sim P_{\text{obj}}(\cdot)}[\mathbb{E}_{m \sim S_{\boldsymbol{\phi}}(\cdot \mid x)}[\log R_{\boldsymbol{\theta}}(x \mid m)] - D_{\text{KL}}(S_{\boldsymbol{\phi}}(M \mid x) || P_\alpha^{\text{prior}}(M \mid \alpha))]$$

$$(\text{By the definition of } \mathcal{J}_{\text{elbo}}(\boldsymbol{\phi}, \boldsymbol{\theta}; \alpha))$$

$$= \mathbb{E}_{x \sim P_{\text{obj}}(\cdot)}[\mathbb{E}_{m \sim S_{\boldsymbol{\phi}}(\cdot \mid x)}[\log R_{\boldsymbol{\theta}}(x \mid m)] - \mathbb{E}_{m \sim S_{\boldsymbol{\phi}}(\cdot \mid x)}[\log S_{\boldsymbol{\phi}}(m \mid x) - \log P_\alpha^{\text{prior}}(m \mid \alpha)]]$$

$$(\text{By the definition of } D_{\text{KL}})$$

$$= \mathbb{E}_{x \sim P_{\text{obj}}(\cdot), m \sim S_{\boldsymbol{\phi}}(\cdot \mid x)}[\log R_{\boldsymbol{\theta}}(x \mid m) + \log P_\alpha^{\text{prior}}(m)] - \mathbb{E}_{x \sim P_{\text{obj}}(\cdot), m \sim S_{\boldsymbol{\phi}}(\cdot \mid x)}[\log S_{\boldsymbol{\phi}}(m \mid x)]$$

$$(\text{By the linearity of } \mathbb{E})$$

$$= \mathbb{E}_{x \sim P_{\text{obj}}(\cdot), m \sim S_{\boldsymbol{\phi}}(\cdot \mid x)}[\log R_{\boldsymbol{\theta}}(x \mid m) + \log P_\alpha^{\text{prior}}(m)] - \mathbb{E}_{x \sim P_{\text{obj}}(\cdot)}[\mathcal{H}(S_{\boldsymbol{\phi}}(M \mid x))]$$

$$(\text{By the definition of } \mathcal{H})$$

$$= \mathbb{E}_{x \sim P_{\text{obj}}(\cdot), m \sim S_{\boldsymbol{\phi}}(\cdot \mid x)}[\log R_{\boldsymbol{\theta}}(x \mid m) - \alpha |m|] + \mathbb{E}_{x \sim P_{\text{obj}}(\cdot)}[\mathcal{H}(S_{\boldsymbol{\phi}}(M \mid x))] - \log Z_\alpha$$

$$(\text{By the definition of } P_\alpha^{\text{prior}})$$

$$= \mathcal{J}_{\text{conv}}(\boldsymbol{\phi}, \boldsymbol{\theta}; \alpha) + \beta \mathbb{E}_{x \sim P_{\text{obj}}(\cdot)}[\mathcal{H}(S_{\boldsymbol{\phi}}(M \mid x))] - \log Z_\alpha.$$

$$(\text{By the definition of } \mathcal{J}_{\text{conv}}(\boldsymbol{\phi}, \boldsymbol{\theta}; \alpha))$$

Thus,

$$\nabla_{\boldsymbol{\phi}, \boldsymbol{\theta}} \mathcal{J}_{\text{elbo}}(\boldsymbol{\phi}, \boldsymbol{\theta}; \alpha)$$
$$= \nabla_{\boldsymbol{\phi}, \boldsymbol{\theta}} \mathcal{J}_{\text{conv}}(\boldsymbol{\phi}, \boldsymbol{\theta}; \alpha) + \nabla_{\boldsymbol{\phi}, \boldsymbol{\theta}} \mathbb{E}_{x \sim P_{\text{obj}}(\cdot)}[\mathcal{H}(S_{\boldsymbol{\phi}}(M \mid x))] - \underbrace{\nabla_{\boldsymbol{\phi}, \boldsymbol{\theta}} \log Z_\alpha}_{=0}$$
$$= \nabla_{\boldsymbol{\phi}, \boldsymbol{\theta}} \mathcal{J}_{\text{conv}}(\boldsymbol{\phi}, \boldsymbol{\theta}; \alpha) + \nabla_{\boldsymbol{\phi}, \boldsymbol{\theta}} \mathbb{E}_{x \sim P_{\text{obj}}(\cdot)}[\mathcal{H}(S_{\boldsymbol{\phi}}(M \mid x))].$$

□

## B.3 CURIOUS CASE ON LENGTH PENALTY

*Remark* 3 (Curious Case on Length Penalty). When $\alpha > \gamma$, $P_\alpha^{\text{prior}}(M)$ converges to a *monkey typing model* $P_{\text{monkey}}^{\text{prior}}(M)$, a unigram model that generates a sequence until emitting eos:

$$\lim_{L_{\max} \to \infty} P_\alpha^{\text{prior}}(m) = P_{\text{monkey}}^{\text{prior}}(m),$$

$$P_{\text{monkey}}^{\text{prior}}(m_t \mid m_{1:t-1}) = P_{\text{monkey}}^{\text{prior}}(m_t) = \begin{cases} 1 - \exp(-\alpha + \gamma) & (m_t = \text{eos}) \\ \exp(-\alpha) & (m_t \neq \text{eos}) \end{cases}, \tag{17}$$

where $\gamma = \log(|\mathcal{A}| - 1) > 0$. Intriguingly, it gives another interpretation of the length penalty. Though originally adopted for modeling sender's *laziness*, the length penalty corresponds to the implicit prior distribution $P_\alpha^{\text{prior}}(m)$, whose special case is monkey typing that totally ignores a context $m_{1:t-1}$. The length penalty may just regularize the policy to a length-insensitive policy rather than a length-sensitive, lazy one.

**Proof of remark 3.** On the one hand, for any message $m = a_1 \cdots a_{|m|} \in \mathcal{M}$, a monkey typing model can be transformed as follows:

$$P_{\text{monkey}}^{\text{prior}}(m)$$

$$= \prod_{t=1}^{|m|} P_{\text{monkey}}^{\text{prior}}(a_t \mid a_{1:t-1}) \qquad \text{(By definition)}$$

$$= P_{\text{monkey}}^{\text{prior}}(\text{eos} \mid a_{1:|m|-1}) \prod_{t=1}^{|m|-1} P_{\text{monkey}}^{\text{prior}}(a_t \mid a_{1:t-1}) \quad \text{(Only the last symbol is eos)}$$

$$= (1 - (|\mathcal{A}| - 1) \exp(-\alpha)) \exp(-\alpha(|m| - 1)) \qquad \text{(By definition)}$$

$$= (\exp(\alpha) - |\mathcal{A}| + 1) \exp(-\alpha|m|).$$

On the other hand, from remark 1, we have:

$$\lim_{L_{\max} \to \infty} Z_\alpha = \lim_{L_{\max} \to \infty} \frac{1 - \{(|\mathcal{A}| - 1) \exp(-\alpha)\}^{L_{\max}}}{\exp(\alpha) - |\mathcal{A}| + 1} = \frac{1}{\exp(\alpha) - |\mathcal{A}| + 1},$$

when $\alpha > \log(|\mathcal{A}| - 1)$. Therefore, we have:

$$\lim_{L_{\max} \to \infty} P_\alpha^{\text{prior}}(m \mid \alpha) = \lim_{L_{\max} \to \infty} \frac{\exp(-\alpha|m|)}{Z_\alpha} = (\exp(\alpha) - |\mathcal{A}| + 1) \exp(-\alpha|m|) = P_{\text{monkey}}^{\text{prior}}(m)$$

when $\alpha > \log(|\mathcal{A}| - 1)$. □

*Remark* 4. In fact, it has been shown (Miller, 1957; Conrad & Mitzenmacher, 2004) that even the monkey typing model $P_{\text{monkey}}^{\text{prior}}(m)$ follows ZLA.

## C  SUPPLEMENTAL INFORMATION ON EXPERIMENTAL SETUP

**Sender Architecture:** The sender $S_\phi$ is based on GRU (Cho et al., 2014) with hidden states of size 512 and has embedding layers converting symbols to 32-dimensional vectors. In addition, LayerNorm (Ba et al., 2016) is applied to the hidden states and embeddings for faster convergence. For $t \geq 0$,

$$
\begin{aligned}
\boldsymbol{e}_{\phi,t} &= \begin{cases} \boldsymbol{\phi}_{\text{bos}} & (t = 0) \\ \text{Embedding}(m_t; \boldsymbol{\phi}_{\text{s-emb}}) & (t > 0) \end{cases}, \\
\boldsymbol{e}_{\phi,t}^{\text{LN}} &= \text{LayerNorm}(\boldsymbol{e}_t), \\
\boldsymbol{h}_{\phi,t} &= \begin{cases} \text{Encoder}(x; \boldsymbol{\phi}_{\text{o-emb}}) & (t = 0) \\ \text{GRU}(\boldsymbol{e}_{t-1}^{\text{LN}}, \boldsymbol{h}_{t-1}^{\text{LN}}; \boldsymbol{\phi}_{\text{gru}}) & (t > 0) \end{cases}, \\
\boldsymbol{h}_{\phi,t}^{\text{LN}} &= \text{LayerNorm}(\boldsymbol{h}_t),
\end{aligned}
\tag{18}
$$

where $\boldsymbol{\phi}_{\text{bos}}, \boldsymbol{\phi}_{\text{s-emb}}, \boldsymbol{\phi}_{\text{o-emb}}, \boldsymbol{\phi}_{\text{gru}} \in \phi$. The object encoder is represented as the additive composition of the embedding layers of attributes. The sender $S_\phi$ also has the linear layer (with bias) to predict the next symbol. For $t \geq 0$,

$$
S_\phi(A_{t+1} \mid m_{1:t}) = \text{Softmax}(\boldsymbol{W}_S \boldsymbol{h}_{\phi,t}^{\text{LN}} + \boldsymbol{b}_S),
\tag{19}
$$

where $\boldsymbol{W}_S, \boldsymbol{b}_S \in \phi$.

**Receiver Architecture:** The receiver has a similar architecture to the sender. The receiver $R_{\boldsymbol{\theta}}$ is based on GRU (Cho et al., 2014) with hidden states of size 512 and has embedding layers converting symbols to 32-dimensional vectors. In addition, LayerNorm (Ba et al., 2016) and Gaussian dropout (Srivastava et al., 2014) are applied to the hidden states and embeddings. The dropout scale is set to 0.001. For $t \geq 0$,

$$
\begin{aligned}
\boldsymbol{e}_{\boldsymbol{\theta},t} &= \begin{cases} \boldsymbol{\theta}_{\text{bos}} & (t = 0) \\ \text{Embedding}(m_t; \boldsymbol{\theta}_{\text{s-emb}}) & (t > 0) \end{cases}, \\
\boldsymbol{e}_{\boldsymbol{\theta},t}^{\text{LN}} &= \text{LayerNorm}(\boldsymbol{e}_t \odot \boldsymbol{d}_e), \\
\boldsymbol{h}_{\boldsymbol{\theta},t} &= \begin{cases} \boldsymbol{0} & (t = 0) \\ \text{GRU}(\boldsymbol{e}_{t-1}^{\text{LN}}, \boldsymbol{h}_{t-1}^{\text{LN}}; \boldsymbol{\theta}_{\text{gru}}) & (t > 0) \end{cases}, \\
\boldsymbol{h}_{\boldsymbol{\theta},t}^{\text{LN}} &= \text{LayerNorm}(\boldsymbol{h}_t \odot \boldsymbol{d}_h),
\end{aligned}
\tag{20}
$$

where $\boldsymbol{\theta}_{\text{bos}}, \boldsymbol{\theta}_{\text{s-emb}}, \boldsymbol{\theta}_{\text{o-emb}}, \boldsymbol{\theta}_{\text{gru}} \in \boldsymbol{\theta}$, $\odot$ is Hadamard product, and $\boldsymbol{d}_e, \boldsymbol{d}_h$ are dropout masks. Note that the dropout masks are fixed over time $t$, following Gal & Ghahramani (2016). The receiver $R_{\boldsymbol{\theta}}$ also has the linear layer (with bias) to predict the next symbol. For $t \geq 0$,

$$
R_{\boldsymbol{\theta}}(A_{t+1} \mid m_{1:t}) = \text{Softmax}(\boldsymbol{W}_R \boldsymbol{h}_{\boldsymbol{\theta},t}^{\text{LN}} + \boldsymbol{b}_R),
\tag{21}
$$

where $\boldsymbol{W}_R, \boldsymbol{b}_R \in \boldsymbol{\theta}$.

**Optimization:** We have to optimize ELBO in which latent variables are discrete sequences, similarly to Mnih & Gregor (2014). By the stochastic computation approach (Schulman et al., 2015), we obtain the following surrogate objective:[11]

$$
\begin{aligned}
\mathcal{J}_{\text{surrogate}}(\boldsymbol{\phi}, \boldsymbol{\theta}) := &\log R_{\boldsymbol{\theta}}(x \mid m) + \beta \sum_{t=1}^{|m|} \log P_{\boldsymbol{\theta}}^{\text{prior}}(m_t \mid m_{1:t-1}) \\
&+ \sum_{t=1}^{|m|} \text{StopGrad}(C_{\boldsymbol{\phi},\boldsymbol{\theta},t} - B_{\boldsymbol{\zeta}}(x, m_{1:t-1}) - b_{\text{mean}}) \log S_\phi(m_t \mid x, m_{1:t-1}),
\end{aligned}
\tag{22}
$$

where $\text{StopGrad}(\cdot)$ is the stop-gradient operator, $B_{\boldsymbol{\zeta}}(x, m_{1:t-1})$ is a state-dependent baseline parametrized by $\boldsymbol{\zeta}$, $b_{\text{mean}}$ is a (state-independent) mean baseline, and

$$
C_{\boldsymbol{\phi},\boldsymbol{\theta},t} = \log R_{\boldsymbol{\theta}}(x \mid m) + \beta \sum_{u=t}^{|m|} (\log P_{\boldsymbol{\theta}}^{\text{prior}}(m_u \mid m_{1:u-1}) - \log S_\phi(m_u \mid x, m_{1:u-1})).
$$

The architecture of the state-dependent baseline $B_{\boldsymbol{\zeta}}(x, m_{1:t-1})$ is defined as:

$$
B_{\boldsymbol{\zeta}}(x, m_{1:t-1}) = B_{\boldsymbol{\zeta}}(\boldsymbol{h}_{\phi,t}^{\text{LN}}) = \boldsymbol{w}_{B,2}^\top \text{LeakyReLU}(\boldsymbol{W}_{B,1} \text{StopGrad}(\boldsymbol{h}_{\phi,t}^{\text{LN}}) + \boldsymbol{b}_{B,1}) + b_{B,2},
$$

---

[11]We mean by "surrogate" that $\nabla_{\boldsymbol{\phi},\boldsymbol{\theta}} \mathcal{J}_{\text{ours}} = \mathbb{E}_{x \sim P_{\text{obj}}(\cdot), m \sim S_\phi(\cdot | x)}[\nabla_{\boldsymbol{\phi},\boldsymbol{\theta}} \mathcal{J}_{\text{surrogate}}]$, i.e., Monte Carlo approximation of $\nabla_{\boldsymbol{\phi},\boldsymbol{\theta}} \mathcal{J}_{\text{ours}}$ is easily performed by using $\mathcal{J}_{\text{surrogate}}$.

where $\boldsymbol{W}_{B,1} \in \mathbb{R}^{256 \times 512}, \boldsymbol{b}_{B,1} \in \mathbb{R}^{256}, \boldsymbol{w}_{B,2} \in \mathbb{R}^{256}, b_{B,2} \in \mathbb{R}$ are learnable parameters in $\boldsymbol{\zeta}$. We optimize the baseline's parameter $\boldsymbol{\zeta}$ by simultaneously minimizing the sum of square errors:

$$\mathcal{J}_{\text{baseline}}(\boldsymbol{\zeta}) := \sum_{t=1}^{|m|} (C_{\boldsymbol{\phi},\boldsymbol{\theta},t} - B_{\boldsymbol{\zeta}}(x, m_{1:t-1}))^2. \tag{23}$$

Note that we optimize the sender's parameter $\phi$ according to $\mathcal{J}_{\text{surrogate}}$ but not $\mathcal{J}_{\text{baseline}}$, so as not to break the original problem setting of ELBO maximization w.r.t $\boldsymbol{\phi}, \boldsymbol{\theta}$. In addition, to further reduce the variance, we use the (state-independent) mean baseline $b_{\text{mean}}$, which is the batch mean of $C_{\boldsymbol{\phi},\boldsymbol{\theta},t} - B_{\boldsymbol{\zeta}}(x, m_{1:t-1})$. We use Adam (Kingma & Ba, 2015) as an optimizer. The learning rate is set to $10^{-4}$. The batch size is set to $8192$. The parameters are updated $20000$ times for each run. To avoid posterior collapse, $\beta$ is initially set $\ll 1$ and annealed to $1$ via the REWO algorithm (Klushyn et al., 2019). We set the constraint parameter of REWO to $0.3$, which intuitively means that it tries to bring $\beta$ closer to $1$ while trying to keep the exponential moving average of the reconstruction error $-\log R_{\boldsymbol{\theta}}(x \mid m)$ less than $0.3$.[12][13]

---

[12]In the conventional setting, Rita et al. (2020) adopted the ad-hoc annealing of the length-penalty coefficient $\alpha$. It might roughly correspond to $\beta$-annealing in VAE.

[13]Our code is available at `https://github.com/mynlp/emecom_SignalingGame_as_betaVAE`.

## D SUPPLEMENTAL INFORMATION ON EXPERIMENTAL RESULT

Figure 3: Results for $n_{\text{bou}}$ (C1), $n_{\text{seg}}$ (C2), $\Delta_{\text{w,c}}$ (C3), C-TopSim (C3), and W-TopSim (C3) are shown in order from the left. The x-axis represents $(n_{\text{att}}, n_{\text{val}})$ while the y-axis represents the values of each metric. The shaded regions and error bars represent the standard error of mean. The threshold parameter is set to $0.25$. The blue plots represent the results for our ELBO-based objective $\mathcal{J}_{\text{ours}}$, the orange ones for (BL1) the conventional objective $\mathcal{J}_{\text{conv}}$ plus the entropy regularizer, and the grey ones for (BL2) the ELBO-based objective whose prior is $P_{\alpha}^{\text{prior}}$. The apparent inferior performance of $\Delta_{\text{w,c}}$ for $\mathcal{J}_{\text{ours}}$ compared to the baselines might be misleading. It is because $\mathcal{J}_{\text{ours}}$ greatly improves both C-TomSim and W-TopSim. The larger scale of their improvements could result in a seemingly worse $\Delta_{\text{w,c}}$, but this does not necessarily indicate poorer performance.

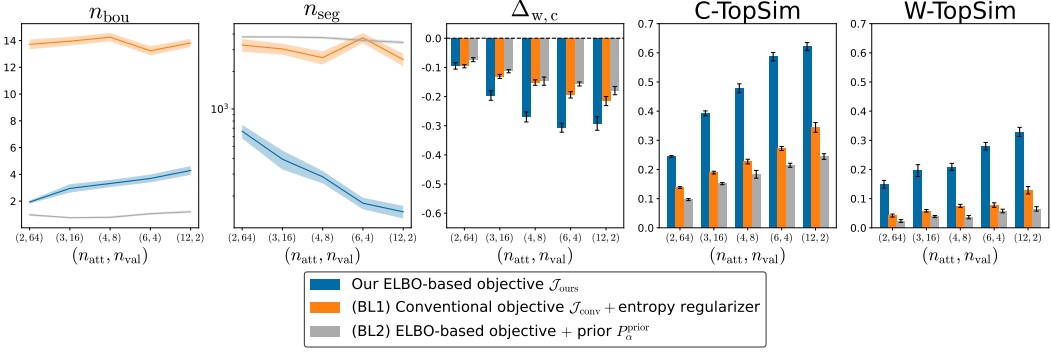

Figure 4: Results for $n_{\text{bou}}$ (C1), $n_{\text{seg}}$ (C2), $\Delta_{\text{w,c}}$ (C3), C-TopSim (C3), and W-TopSim (C3) are shown in order from the left. The x-axis represents $(n_{\text{att}}, n_{\text{val}})$ while the y-axis represents the values of each metric. The shaded regions and error bars represent the standard error of mean. The threshold parameter is set to $0.25$. The blue plots represent the results for our ELBO-based objective $\mathcal{J}_{\text{ours}}$, the orange ones for (BL1) the conventional objective $\mathcal{J}_{\text{conv}}$ plus the entropy regularizer, and the grey ones for (BL2) the ELBO-based objective whose prior is $P_{\alpha}^{\text{prior}}$. The apparent inferior performance of $\Delta_{\text{w,c}}$ for $\mathcal{J}_{\text{ours}}$ compared to the baselines might be misleading. It is because $\mathcal{J}_{\text{ours}}$ greatly improves both C-TomSim and W-TopSim. The larger scale of their improvements could result in a seemingly worse $\Delta_{\text{w,c}}$, but this does not necessarily indicate poorer performance.

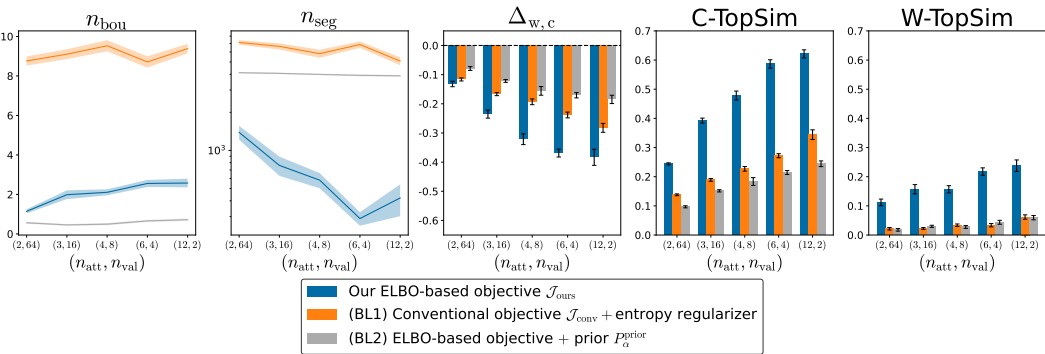

We show the supplemental results in Figure 3 and Figure 4, in which $n_{\text{bou}}$ (C1), $n_{\text{seg}}$ (C2), $\Delta_{\text{w,c}}$ (C3), C-TopSim (C3), and W-TopSim (C3) are shown in order from the left. While Figure 2 in the main content shows the result for threshold $= 0$, Figure 3 shows the result for threshold $= 0.25$ and Figure 4 for threshold $= 0.5$. In Figure 4, the blue line plotted for $n_{\text{seg}}$ does not decrease monotonically, i.e., C2 is a bit violated when using our objective $\mathcal{J}_{\text{ours}}$ as well as the baseline objectives. It implies that the threshold parameter $0.5$ is not appropriate in our case.

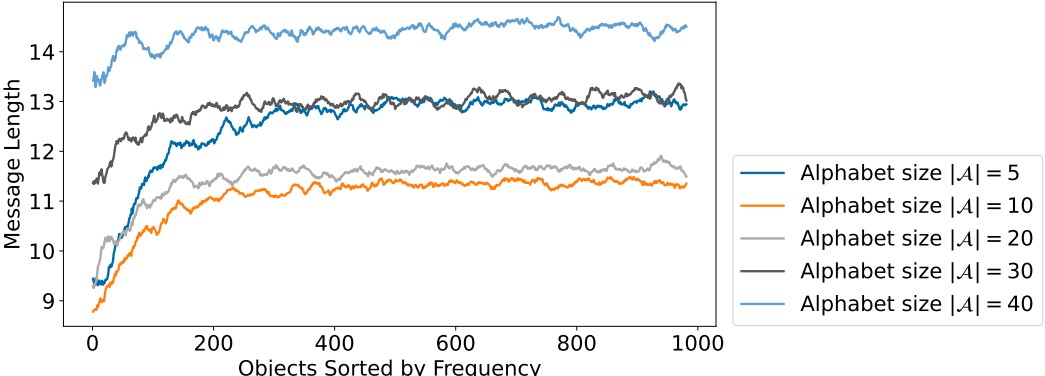

Figure 5: Mean message length sorted by objects' frequency across 32 random seeds. A moving average with a window size of 10 is shown for readability.

## E  SUPPLEMENTAL EXPERIMENT ON ZIPF'S LAW OF ABBREVIATION

### E.1  SETUP

**Object and Message Space:** We set $\mathcal{X} := \{1, \ldots, 1000\}$, $P_{\mathrm{obj}}(x) \propto x^{-1}$, $L_{\mathrm{max}} = 30$, and $|\mathcal{A}| \in \{5, 10, 20, 30, 40\}$ following (Chaabouni et al., 2019).

**Agent Architectures and Optimization:** The architectures of sender $S_\phi$ and $R_\theta$ are the same as described in Section 4.1. The optimization method and parameters are also the same as described in Section 4.1.

### E.2  RESULT AND DISCUSSION

The experimental results are shown in Figure 5. As can be seen from the figure, there is a tendency for shorter messages to be assigned to high-frequency objects.

It is also observed that the larger the alphabet size $|\mathcal{A}|$, the longer the messages tend to be. It is probably because, as the alphabet size $|\mathcal{A}|$ increases, the likelihood of `eos` becomes smaller in the early stages of training.

As an intuitive analogy for this, let us assume that a sender $S_\phi(M|X)$ and prior message distribution $P_\theta^{\mathrm{prior}}(M)$ are roughly similar to a *uniform monkey typing* (UMT) in the early stages of training:

$$S_\phi(m_t \mid x, m_{1:t-1}) \approx \frac{1}{|\mathcal{A}|}, \quad P_\theta^{\mathrm{prior}}(m_t \mid m_{1:t-1}) \approx \frac{1}{|\mathcal{A}|}. \tag{24}$$

The length distribution of UMT is a *geometry distribution* with a parameter $p = |\mathcal{A}|^{-1}$ (Imagine a situation where you roll the dice with "`eos`" written on one face until you get that face). Then, the expected message length of UMT is $p^{-1} = |\mathcal{A}|$. Therefore, as the alphabet size $|\mathcal{A}|$ increases, the sender $S_\phi(M|X)$ and prior message distribution $P_\theta^{\mathrm{prior}}(M)$ are inclined to prefer longer messages in the early stages. Consequently, they are more likely to converge to longer messages. Designing neural network architectures that maintain a consistent scale for `eos` regardless of alphabet size is a challenge for future work.

## F   SUPPLEMENTAL INFORMATION FOR RELATED WORK

Table 2: We show the characteristics of this paper and related EC work in several respects to clarify the positioning of this paper. We mean by "**Generative?**" whether their formulation is based on a generative perspective such as VAE and VIB, by "**Variable message length?**" whether message length in their settings is variable, by "**Compositionality?**" whether they study the compositionality (e.g., TopSim) of emergent languages, by "**ZLA?**" whether they investigate if emergent languages follow ZLA, by "**HAS?**" whether they investigate if emergent languages follow HAS, by "**Prior?**" whether they are aware of the existence of prior distributions and carefully choose an appropriate one. The check mark (✓) indicates yes, and the blank indicates no.

| Paper | Generative? | Variable message length? | Compositionality? | ZLA? | HAS? | Prior? |
|---|:---:|:---:|:---:|:---:|:---:|:---:|
| Chaabouni et al. (2021) Tucker et al. (2022) | ✓ | | | | | |
| Chaabouni et al. (2019) Rita et al. (2020) Ueda & Washio (2021) | | ✓ | | ✓ | | |
| Andreas (2019) Li & Bowling (2019) Chaabouni et al. (2020) Ren et al. (2020) | | | ✓ | | | |
| Resnick et al. (2020) | ✓ | | ✓ | | | |
| Taniguchi et al. (2022) Inukai et al. (2023) | ✓ | | | | | |
| Ueda et al. (2023) | | | ✓ | ✓ | ✓ | |
| Ours | ✓ | ✓ | ✓ | ✓ | ✓ | ✓ |

