# OpenReview forum: "Lewis's Signaling Game as beta-VAE For Natural Word Lengths and Segments"
_ICLR.cc/2024/Conference — ICLR 2024 poster_

### Official Review · Reviewer_TgUR · 2023-10-27

**Soundness:** 3 good
**Presentation:** 2 fair
**Contribution:** 2 fair
**Rating:** 5
**Confidence:** 3

**Summary:**

The authors discuss connections between classic emergent communication in Lewis signalling games and Beta-VAEs.

In some traditional EC works, a speaker and listener must coordinate such that the listener can reconstruction a speaker's "target" observation, given communication. In many ways, this mirrors classic reconstruction training. Prior works have often found that the resulting communication from such training is often "unhumanlike" in several ways, including ZLA and HAS metrics. This work argues that such undesireable properties are likely a result of implicit priors that most EC works encode.

By connecting EC to Beta-VAE methods, the authors uncover theoretical interpretations of different terms in EC and open up the important directions for experiments (such as varying prior distributions or Beta).

In experiments, the authors show that, by using a learnable prior in training agents, they appear to achieve greater separation of EC into "word-like" units.

**Strengths:**

## Originality
I'm am somewhat torn about the originality of this work. On the one hand, I think connection EC literature to other theoretically-rich approaches like beta-VAE is a very good idea. On the other hand, the authors note that some prior literature appears to have considered a generalization of this problem ("Section 6: Tucker et al. defined a communication game... based on VIB, which is known as a generalization of beta-VAE."), which makes me think this work is not proposing novel ideas.

## Quality
Overall, the work seems quite careful and sound in discussing the mathematical underpinnings of many EC methods.

## Clarity
Some aspects of this paper were quite clear (in particular, the introduction and conclusion are very good), but I found other aspects harder to read. I suspect this is somewhat due to having a fair amount of notation is not immediately interpretable without remembering definitions from earlier pages (e.g., "monkey typing model" or n_bou).

## Significance
I think this work falls within an important (and significant) field of connecting EC to other training methods and objects. I remain somewhat confused about the relationship to prior art, however, so I am unsure of the significance of this individual work.

**Weaknesses:**

Overall, I like aspects of this work, but there are a few important unresolved questions or weaknesses that I would want to see addressed before accepting, in particular about relations to prior art.

## Relation to prior art
The authors do a good job noting related prior literature, but I remain somewhat confused by the position of this paper relative to such literature. In particular, the authors write that:

> Moreover, Resnick et al. (2020) explicitly formulated the objective as ELBO, though it is not directly applicable to this paper...

> Tucker et al. (2022) defined a communication game called VQ-VIB, based on Variational Information Bottleneck (VIB, Alemi et al., 2017) which is known as a generalization of beta-VAE. Also, Chaabouni et al. (2021) formalized a color naming game with a similar motivation.

If prior art has used the same formulation and considered a generalization of the problem this paper is considering, what are the contributions of this paper? Honestly, I suspect there are many unique contributions made in this paper, but the contrast relative to prior art should be made much more obvious. Even just adding a sentence at the end of each related works section saying, e.g., "While Tucker et al., Alemi et al., and Chaabouni et al. consider similar frameworks to us, we introduce novel metrics and results" or something to that effect would help a lot. Ideally, the authors would run experiments comparing to Resnick's method.

## Presentation of results
I found the results somewhat difficult to read. Figure 2 contains the main results, and with enough flipping between pages, I could eventually figure out how to interpret them, but generally I encourage authors to make figures more self-contained. For example, listing a baseline as BL1 is not as informative as using a name/label that actually describes characteristics of the baseline (e.g., conventional + entropy).

## Why did segments become more meaningful.

The analysis in this section, while addressing a very important question, is slightly unsatisfying. First, parts of the writing are very casual (e.g., The receiver must be surprised several times..."), whereas in reality the receiver just needs to receive, over multiple timesteps, enough bits to reconstruct the input. It is unclear what it means to "be surprised" as a binary term.

Second, I question the fundamental conclusion of this paragraph. The authors appear to suggest that the competing terms for entropy and reconstruction are what give rise to word boundaries. In other words, communication wants to often be predictable (because of the entropy term), which creates word-like clumps. However, as the authors note, the speaker needs to communicate some information in at least some timesteps to convey the meaning to the listener. Is there any mathematical basis, given the training terms used, for why that information should be concentrated in just a few timesteps (which would match word-like clumps) as opposed to evenly distributed across time? For example, in a simple four-timestep case conveying 4 bits, is there any advantage (as measured by decreased loss) to transmitting [2 bits, 0 bits, 2 bits, 0 bits] vs. [1 bit, 1 bit, 1 bit, 1 bit]?

## Minor:
Appendix D would greatly benefit from a little bit more text explaining what the graphs present. There are also some sentences that need editing (e.g., "threshold is set to 0.25.")

**Questions:**

1. In the Weaknesses section, I raised questions about why an entropy term would actually increase word segmentation. To repeat it here: is there any mathematical reason that the losses used during training should concentrate surprisal in just a few timesteps (which would induce word-like clumps) instead of spreading the surprisal loss more evenly across time?

2. I struggled to understand Figure 5. What is it depicting? What do the legend entries/different lines correspond to?

3. Just a clarifying question about the results for Criterion C3: the authors' proposed method is worse than baselines, correct? I recognize that topsim values for the proposed method improved generally, but for the narrow metric of the difference between topsim values, there is a decrease, right?

---

> ### Author Response · Authors · 2023-11-17
> **Response to Reviewer TgUR (Comment 1-2)**
>
> Thank you very much for your detailed and thoughtful reviews.
> We have extensively revised our paper in light of the feedback received.
> These revisions aim to clarify the paper's positioning and enhance the readability of our experimental results.
>
> ## Reviewer's Comment 1
> > The authors do a good job noting related prior literature, but I remain somewhat confused by the position of this paper relative to such literature. ...
>
> Thank you for point it out.
> We added the positioning of our paper, compared to the related studies Resnick et al. (2020); Chaabouni et al. (2021); Tucker et al. (2022).
>
> Perhaps another problem was the beginning of Sec 3: `The main proposal of this paper is redefine the objective of signaling game as (beta-)VAE's`.
> We modified the beginning of Sec 3 too.
>
> Moreover, we made the table to clarify our positioning, which is put to Appendix F in the revised version.
>
> ## Revision for Comment 1
> At Paragraph "EC as Representation Learning" in Sec 6 "Related Work", we added:
> > (***Revision***) Resnick et al. (2020) formulated the objective as ELBO. However, they defined messages as fixed-length binaries and agents as non-autoregressive models, which is not applicable to our variable-length setting. Moreover, they do not discuss the prior distribution choice; they seem to have adopted Bernoulli purely for computational convenience. In contrast, we indicated that their choice influences the structure of emergent languages.
>
> At Pragraph "VIB" in Sec 6 "Related Work", we added:
> > (***Revision***) Note that messages are defined as single symbols in their settings. In contrast, messages are of variable length in our setting to discuss the structure of emergent languages such as ZLA and HAS.
>
> In Sec 3, we modified the first few sentences:
> > (***Revision***) ~~The main proposal of this paper is to redefine the objective of signaling games as (beta-)VAE's, i.e., the evidence lower bound (ELBO) with an additional KL weighting hyperparameter $\beta$.~~
> > In this section, we first redefine the objective of signaling games as (beta-)VAE's, i.e., the evidence lower bound (ELBO) with a KL weighting hyperparameter $\beta$. Next, we show several reasons supporting our ELBO-based formulation.
>
> In Appendix F, we added the table to clarify our positioning.
> The table shows the characteristics of the studies in terms of:
> - whether their formulation is based on a generative perspective such VAE and VIB,
> - whether message length in their settings is variable,
> - whether they study the compositionality of emergent languages,
> - whether they investigate if emergent languages follow ZLA,
> - whether they investigate if emergent languages follow HAS, and
> - whether they are aware of the existence of prior distributions and carefully choose an appropriate one.
>
> Each related study stasfies some of them, while it does not satisfy the others.
> In contrast, our paper satisfies all of them.
>
> ## Reviewer's Comment 2
> > I found the results somewhat difficult to read. Figure 2 contains the main results, and with enough flipping between pages, I could eventually figure out how to interpret them, but generally I encourage authors to make figures moreself-contained. For example, listing a baseline as BL1 is not as informative as using a name/label that actually describescharacteristics of the baseline (e.g., conventional + entropy).
> > ...
> > ...
> > Appendix D would greatly benefit from a little bit more text explaining what the graphs present. There are also some sentences that need editing (e.g., "threshold is set to 0.25.")
>
> Thank you for this feedback.
>
> ## Revision for Comment 2
> To make Figure 2; 3; 4 more readable and self-contained, we added more information to the legend and included the explanation on the evaluation metrics and the baseline methods in the captions.
> As a result, the caption (in Figure 2) became as follows:
> > (***Revision***) Results for $n_{\textrm{bou}}$ (C1), $n_{\textrm{seg}}$ (C2), $\Delta_{w,c}$ (C3), C-TopSim (C3), and W-TopSim (C3) are shown in order from the left. The x-axis represents $(n_{\textrm{att}},n_{\textrm{val}})$ while the y-axis represents the values of each metric. The shaded regions and error bars represent the standard error of mean. The threshold parameter is set to $0$. The blue plots represent the results for our ELBO-based objective $\mathop{\mathcal{J}}\nolimits_{\textrm{ours}}$, the orange ones for (BL1) the conventional objective $\mathop{\mathcal{J}}\nolimits_{\textrm{conv}}$ plus the entropy regularizer, and the grey ones for (BL2) the ELBO-based objective whose prior is $P_{\alpha}^{\textrm{prior}}$.
> > ...... (*For the rest part of the revised caption, see our response to Comment 6*)

---

> > ### Author Response · Authors · 2023-11-17
> > **Response to Reviewer TgUR (Comment 3-5)**
> >
> > ## Reviewer's Comment 3
> > > The analysis in this section, while addressing a very important question, is slightly unsatisfying.
> > > First, parts of thewriting are very casual (e.g., The receiver must be surprised several times..."), whereas in reality the receiver just needs to receive, over multiple timesteps, enough bits to reconstruct the input. It is unclear what it means to "be surprised"as a binary term.
> >
> > Thank you for pointing that out.
> >
> > ## Revision for Comment 3
> > To make the discussion more mathematically grounded, we added some mathematical terms, after casually used "surprised", as follows:
> > > (***Revision***) a receiver has to be "surprised," i.e., to receive symbols $m_{t}$ with relatively high $-\log P_{\theta}(m_{t}|m_{1:t-1})$.
> >
> > ## Reviewer's Comment 4
> > > Second, I question the fundamental conclusion of this paragraph. The authors appear to suggest that the competingterms for entropy and reconstruction are what give rise to word boundaries. In other words, communication wants tooften be predictable (because of the entropy term), which creates word-like clumps. However, as the authors note, the speaker needs to communicate some information in at least some timesteps to convey the meaning to the listener. Is there any mathematical basis, given the training terms used, for why that information should be concentrated in just a few timesteps (which would match word-like clumps) as opposed to evenly distributed across time? For example, in a simple four-timestep case conveying 4 bits, is there any advantage (as measured by decreased loss) to transmitting [2bits, 0 bits, 2 bits, 0 bits] vs. [1 bit, 1 bit, 1 bit, 1 bit]?
> > > ...
> > > ...
> > > In the Weaknesses section, I raised questions about why an entropy term would actually increase word segmentation. To repeat it here: is there any mathematical reason that the losses used during training should concentrate surprisal in just a few timesteps (which would induce word-like clumps) instead of spreading the surprisal loss more evenly across time?
> >
> > Thank you for your insightful comment, which has provided valuable perspectives to our discussion.
> > Your feedback prompts us to think more deeply about the issue.
> >
> > Currently we have the following hypothesis, though it's more like an intuition, rather than a mathematical basis:
> > - Agents themselves do not necessarily have any strong motivation to make bit pattern zigzag or flat.
> > - However, as a matter of probability, it is simply difficult to converge to a flat one among infinitely many possible bit patterns (almost all of them would zigzag).
> > - The bit patterns are optimized under the trade-off between reconstruction and surprisal, while they (almost surely) fail to find a flat pattern.
> >
> > ## Revision for Comment 4
> > To avoid overstatement, we slightly weakened our statement in discussion, as follows:
> > > (***Revision***) Though it is not straightforward to give a clear mathematical basis, we currently hypothesize that the improvement of segments' meaningfulness is due to the competing pressure of the reconstruction and the surprisal.
> >
> > Also, to clarify the limitation of our discussion, we added a footnote to mention that our current hypothesis does not fully explain why the bit pattern does not converges to a flat one, as follows:
> > > (***Revision***) Our current hypothesis does not fully explain the reason why the branching entropy does not converge to a flat one instead of a jittering one. At the moment, we conjecture that, as a matter of probability, it is simply difficult to converge to a flat one among infinitely many possible bit patterns (almost all of them would jitter).
> >
> > ## Reviewer's Comment 5
> > >  I struggled to understand Figure 5. What is it depicting? What do the legend entries/different lines correspond to?
> >
> > We appreciate your feedback regarding the lack of legend descriptions in Figure 5.
> >
> > ## Revision for Comment  5
> > To make them understandable, we explictly describe what the legends mean, e.g., "Alphabet size $|\mathcal{A}|=5$".

---

> > > ### Author Response · Authors · 2023-11-17
> > > **Response to Reviewer TgUR (Comment 6)**
> > >
> > > ## Reviewer's Comment 6
> > > > Just a clarifying question about the results for Criterion C3: the authors' proposed method is worse than baselines,correct? I recognize that topsim values for the proposed method improved generally, but for the narrow metric of the difference between topsim values, there is a decrease, right?
> > >
> > > Thank you for another very insightful comment.
> > >
> > > We understand your concern, but we would like to claim that it is safer not to explicitly state that it is better/worse than baselines.
> > >
> > > This is because the following interpretation is also possible: "Relatively large (resp. small) Score X minus relatively large (resp. small) Score Y would simply yield a relatively large (resp. small) difference (X=W-TopSim and Y=C-TopSim in our case).
> > > In other words,  it may simply be a matter of scale.
> > >
> > > In our opinion, what we can take from the result of $\Delta_{w,c}$ is only whether it is positive (W-TopSim>C-TopSim), negative (W-TopSim<C-TopSim), or zero (W-TopSim=C-TopSim).
> > >
> > > ## Revision for Comment 6
> > > Nevertheless, future readers may have the same impression.
> > > For clarity, we briefly mention that point in the captions of Figure 2; 3; 4, as follows:
> > > > (***Revision***) ...... The apparent inferior performance of $\Delta_{w,c}$ for $\mathop{\mathcal{J}}\nolimits_{\textrm{ours}}$ compared to the baselines might be misleading. It is because $\mathop{\mathcal{J}}\nolimits_{\textrm{ours}}$ greatly improves both C-TomSim and W-TopSim. The larger scale of their improvements could result in a seemingly worse $\Delta_{w,c}$, but this does not necessarily indicate poorer performance.

---

> > > ### Comment · Reviewer_TgUR · 2023-11-22
> > >
> > > Thank you for your extensive comments (here and elsewhere).
> > >
> > > I am glad that we seem to agree about comment 4 (that there is no fundamental reason for increasing HAS). Weakening the statements as you have proposed addresses this concern, but I feel like this fact weakens the strength of the paper overall. In other words, given that the authors appear to state that there is not theoretical reason that their reframing should increase HAS, such effects are unmotivated/purely empirical. That is fine but undercuts the main point of the paper as a theoretical re-framing of EC as Beta-VAE.

---

### Official Review · Reviewer_E7HT · 2023-10-28

**Soundness:** 3 good
**Presentation:** 3 good
**Contribution:** 3 good
**Rating:** 8
**Confidence:** 3

**Summary:**

This paper reanalyzes an emergent communication-signalling game in terms of
a VAE.  Within this analysis, the optimization of a signalling game is using an
"implicit prior" which leads to statistical properties of the emerging language
which do not match human languages.  Introducing linguistically-inspired priors
into signalling game by way of the VAE framework improves the resulting
emergent languages' statistical properties (i.e., adhering Zipf's Law of
Abbreviation and Harris's Articulation Scheme more closely).

**Strengths:**

- (major) The paper aims at re-analyzing a common EC setting in a more
  formalized way, yielding the potential for theoretical insights that would
  not otherwise be possible.
- (major) Furthermore, I think this analysis is largely in the correct
  direction with analyzing the signalling game as a VAE, looking at inductive
  biases, and tying in linguistic concepts like Zipf's Law of Abbreviation and
  Harris's Articulation Scheme (although this wide scope is also a bit of
  concern; cf. "Weaknesses").
- (minor) The experiments partially satisfy HAS which is known to hold for
  human languages.

**Weaknesses:**

- (major) A critical part of the paper is the "prior" within a VAE or
  signalling game, but I did not get a concrete sense of what this prior
  actually is in the context of a signalling game with neural network-based
  agents (I expand on this in "Questions").  As a result, it makes me unsure
  how well the theoretical claims actually apply to a real setup.
- (major) The paper, I think, tried to do too much, and ends up not spending
  enough time on the core claims, namely, the signalling game can be
  re-analyzed as a VAE.  I think the paper would benefit greatly from cutting
  away all but the essential claims and going through those more slowly and
  thoroughly.
    - For example, this shows up in the experiments which seem more concerned
      with evaluating the existence of ZLA/HAS in the newly proposed setting
      rather than establishing empirically that the signalling game behaves
      like a VAE.
- (minor) The notation and the proofs are not very clear, and it made it
  slow/difficult to work through the equations.

**Questions:**

- What exactly is the "prior" in the emergent language game?  I understand that
  it is implicit, but does that mean that is embedded in the objective function
  (i.e., the $D_\text{KL}$ term is constant)?  Or is it instead the case that
  the sender's architectural biases represent the prior?
- In addition to the theoretical analysis, what else can the authors point to
  to support the claim that an EC signalling game is analogous to a VAE?


### Other comments

- If the authors are assuming a REINFORCE objective for the signalling game,
  that should be mentioned earlier than Sec 4.1.
- What is $A_t$ in Eq 3?
- In Sec 2.2, I do not think the section compositionality is relevant or
  important; it should removed, in that case.
- Before Eq 5, what is $\mathcal A^*$?  Is it supposed to be a Kleene star?
- What exactly is the uniform prior?  Is it just a constant probability mass
  over every possible sequence?  If so, how do we know that is the "implicit
  prior" and not something like a uniform _unigram_ prior instead, for example.


- Sec 3.1: It is not clear to me how (9) is derived from (2).  I looked at Section
  B.1, but it was very unclear what was happening because rather than starting
  with (2) and going to (9), it talks about "transforming" different sides of
  the equation.
  - It would also be helpful to give an indication of what from Schulman et al.
    (2015) is being applied (i.e., the what the "stochastic computation
    approach" is).
  - As a result, I'm not convinced that the reconstruction game, absent
    modifications to the traditional object (e.g., length penalty), assumes
    a uniform prior of messages.
  - It seems like the $P(m) = \mathbb E_{x\sim{}P_\text{obj}}[S(m|x)]$ should be the
    prior over messages.  I very well might be misunderstanding something here
    due to terminology.  Am I conflating here that "prior" as the distribution
    of messages the receiver produces given the distribution over inputs with
    "prior" in the sense of our objective function which we are optimizing
    against (in which case "prior" does not refer to anything concrete in the
    EC environment but rather only to the optimization process by analogy to
    a VAE's optimization)?
- Sec 3.2: what is a "heuristic variant of [a] VAE"?
- Sec 3.4:
  - The very first paragraph of this section, I think, is glossing over
    critical question in the paper: what is the connection between the "prior"
    and the actual EC setup.  I understand that the EC setup is analogous to
    a VAE, but what exactly is the analog of the VAE's prior?
  - I think the use of "approximately" is dangerous when trying to make
    theoretical claims; I understand that it is unavoidable in something as
    messy as EC, but it still needs to be accompanied by some justification in
    order to keep the theoretical claims strong.


### Minor notes

- "they are not" reproduced emergent lanuaguages   has awkward phrasing
- Right after Eq. (1), it should be $\log(|\mathcal A| - 1)\ge0$ in the case
  that $|\mathcal A| = 2$.
- Sec 3.3:
  - what is $\mathcal M$ -- the set of all messages?
  - What is $\mathcal A$, again?
  - Eq 15 limit notation here would be more appropriate

---

> ### Author Response · Authors · 2023-11-17
> **Response to Reviewer E7HT (Comment 1-4)**
>
> Thank you for your thorough and insightful review.
> In response to the comments provided, we have diligently revised our paper.
> Your constructive feedback has been instrumental in enhancing the readability and academic value of our work.
>
> ## Reviewer's Comment 1
> > (major) A critical part of the paper is the "prior" within a VAE or signalling game, but I did not get a concrete sense of what this prior actually is in the context of a signalling game with neural network-based agents (I expand on this in "Questions"). As a result, it makes me unsure how well the theoretical claims actually apply to a real setup.
>
> We greatly appreciate your feedback regarding the prior in the context of EC signalling game.
> (Please see our response to Comment 3; 9; 10; 11; 12)
>
> ## Reviewer's Comment 2
> > (major) The paper, I think, tried to do too much, and ends up not spending enough time on the core claims, namely, the signalling game can be re-analyzed as a VAE. I think the paper would benefit greatly from cutting away all but the essential claims and going through those more slowly and thoroughly.
>
> We value your suggestion to focus more on the core claims of our paper.
> In the revised version, we have taken the reviewers' comments into consideration and made efforts to improve the quality of the paper.
> We have removed unnecessary text and moved some parts to the appendix to focus more on the essential aspects.
> We hope you will find the revised version more satisfactory.
>
> Roughly speaking, our core claims are made of two components, which are (somewhat philosophically) of mutual dependence.
>
> Our core claims are:
> 1. The conventional EC signaling game is analogous to (beta-)VAE.
> 2. By explicitly re-defining EC signaling game as (beta-)VAE, the structure of emergent languages improves.
>
> Thanks to the 2nd claim,
> - the 1st claim becomes more than a mere theoretical result.
>
> Thanks to the 1st claim,
> - the 2nd claim becomes more than an arbitrary "hack" of the evaluation criteria (The arbitrary hack might be a questionable practice as a simulation of language emergence, in a scientific sense).
>
> For a more detailed decomposition of our claims, please refer to our response to Comment 4.
>
> ## Reviewer's Comment 3
> > What exactly is the "prior" in the emergent language game? I understand that it is implicit, but does that mean that is embedded in the objective function (i.e., the term is constant)? Or is it instead the case that the sender's architectural biases represent the prior?
>
> We appreciate your question regarding what exactly the prior is.
> This is an important point to clarify our storytelling.
>
> In terms of the conventional EC signaling game, yes, the prior is constant.
> Specifically, it is a uniform distribution over a message space.
>
> (In contrast, we regard it as a language model)
> (See also our response to Comment 9; 10; 11; 12).
>
> ## Reviewer's Comment 4
> > In addition to the theoretical analysis, what else can the authors point to to support the claim that an EC signalling game is analogous to a VAE?
>
> Thank you for questioning that point.
> Our contribution is indeed more than the theoretical analysis of the game, including e.g.,
> - a new idea for EC setting (i.e., prior as a language model),
> - empirical results (for ZLA and HAS), and
> - relationship to the linguistic concept (i.e., surprisal theory).
>
> Our overall claims are as follows:
>
> First, as you mentioned,
> - ***As an interesting math problem***, EC signaling game is similar to (beta-)VAE (Sec 3.1; 3.2).
>
> Importantly,
> - By interpreting EC signaling game as (beta-)VAE, ***we can reveal the existence of implicit prior*** (Sec 3.1).
>
> Furthermore, by having revealed the existence of implicit prior,
> - We can appropriately re-define the prior distribution to have a ***good influence on the structure of emergent languages***, namely ZLA/HAS (Sec 3.3; 3.4; Sec 4).
> - We can naturally introduce a ***language model (=prior distribution) explicitly in the simulation of language emergence*** (Sec 3.4).
>
> Moreover, by regarding the prior as a language model,
> - We'll be somewhat ***satisfied in a scientific sense*** to notice that ***the prior coincides with surprisal, a concept from psycho-linguistics*** (Sec 3.5).
>
> They are also shown in "Recipe for Supporting Our ELBO-based Formulations" in Sec 3.

---

> > ### Author Response · Authors · 2023-11-17
> > **Response to Reviewer E7HT (Comment 5-10)**
> >
> > ## Reviewer's Comment 5 (about Sec 4.1)
> > > If the authors are assuming a REINFORCE objective for the signalling game, that should be mentioned earlier than Sec 4.1.
> >
> > Thank you for pointing it out.
> > We agree with your point.
> >
> > ## Revision for Comment 5
> > In Sec 2, we added the following explanation on the gradient of the (conventional) objective function, which naturally indicates we consider REINFORCE-like optimization in this paper.
> > > (***Revision***) The gradient of $\mathop{\mathcal{J}}\nolimits_{\textrm{conv}}(\phi,\theta)$ is obtained as (Chaabouni et al. 2019): $$\nabla_{\phi,\theta}\mathop{\mathcal{J}}\nolimits_{\textrm{conv}}(\phi,\theta)=\mathop{\mathbb{E}}\nolimits_{x\sim P_{\textrm{obj}}(\cdot),m\sim S_{\phi}(\cdot|x)}\left[\nabla_{\phi,\theta}\log R_{\theta}(x|m)+(\log R_{\theta}(x|m)-B(x))\nabla_{\phi,\theta}\log S_{\phi}(m|x)\right],$$ where $B:\mathcal{X}\to\mathbb{R}$ is a baseline. Intuitively, $\phi$ is updated via REINFORCE (Williams, 1992) and $\theta$ via the standard backpropagation.
> >
> > ## Reviewer's Comment 6 (about Sec 2.1)
> > > What is $A_{t}$ in Eq 3?
> >
> > We meant by $A_{t}$ a variable of a finite alphabet $\mathcal{A}$.
> >
> > ## Revision for Comment 6
> > To make it more direct and natural, we rewrote $A_{t}$ to $M_{t}$ ($t$-th message symbol variable).
> > > (***Revision***) $$\nabla_{\phi,\theta}^{\textrm{entreg}}:=\frac{\lambda^{\text{entreg}}}{|m|}\sum\nolimits_{t=1}^{|m|}\nabla_{\phi,\theta}\mathcal{H}(S_{\phi}(M_{t}|x,m_{1:t-1})),\tag{3}$$
> >
> > ## Reviewer's Comment 7 (about Sec 2.2)
> > > In Sec 2.2, I do not think the section compositionality is relevant or important; it should removed, in that case.
> >
> > Thank you for the feedback.
> > We agree with your point.
> >
> > ## Revision for Comment 7
> > We moved the section "Compositionality" to the footnote.
> >
> > ## Reviewer's Comment 8 (about Sec 2.2)
> > > Before Eq 5, what is $A^{*}$? Is it supposed to be a Kleene star?
> >
> > Yes, we meant by $\mathcal{A}^{*}$ a Kleene star.
> >
> > ## Revision for Comment 8
> > To make it more intuitive, we use a more direct notation instead of the Kleene star.
> > > (***Revision***) $s=a_{1}\cdots a_{n}$ ($a_{i}\in\mathcal{A}$)
> >
> > ## Reviewer's Comment 9 (about Sec 3.1)
> > > What exactly is the uniform prior? Is it just a constant probability mass over every possible sequence? If so, how do we know that is the "implicit prior" and not something like a uniform $unigram$ prior instead, for example.
> >
> > Yes, the former is the case.
> > The implicit prior $P_{\textrm{unif}}^{\textrm{prior}}$ is a uniform distribution over a finite message space (constant w.r.t messages $m$).
> > $P_{\textrm{unif}}^{\textrm{prior}}(m)$ is not a unigram (see the example below).
> >
> > **Example:**
> > Let a finite alphabet be $\mathcal{A}:=\\{\texttt{eos},0,1\\}$ and a message space be $\mathcal{M}$.
> > Consider two messages: $m^{(1)}=0101\texttt{eos}$ and $m^{(2)}=01010101\texttt{eos}$.
> > If we consider a (uniform) unigram model, then $P(m^{(1)})=(\frac{1}{3})^{5}$ and $P(m^{(2)})=(\frac{1}{3})^{9}$, which are not the same.
> > In contrast, if we consider literary a uniform distribution, then $P(m^{(1)})=P(m^{(2)})=\frac{1}{|\mathcal{M}|}$.
> >
> > ## Revision for Comment 9
> > To make it more direct and clear, we rewote $P_{\textrm{unif}}^{\textrm{prior}}(m)\propto\textrm{const}$ to $P_{\textrm{unif}}^{\textrm{prior}}(m)=\frac{1}{|\mathcal{M}|}$.
> > > (***Revision***) $P_{\textrm{unif}}^{\textrm{prior}}(m)=\frac{1}{|\mathcal{M}|}$
> >
> > ## Reviewer's Comment 10 (about Sec 3.1; Appendix B.1)
> > > Sec 3.1: It is not clear to me how (9) is derived from (2). I looked at Section B.1, but it was very unclear what was happening because rather than starting with (2) and going to (9), it talks about "transforming" different sides of the equation.
> >
> > As you point out, our proof is unclear.
> > We made the following revision.
> > We hope that it contributes to the interpretability and readability of our proof.
> >
> > ## Revision for Comment 10
> > - To clarify the proof, we briefly recall the definition of the target equation and relevant terms.
> > - To clarify the proof, we briefly explain what we will transform.

---

> ### Author Response · Authors · 2023-11-17
> **Response to Reviewer E7HT (Comment 11-13)**
>
> ## Reviewer's Comment 11 (about Sec 3.1; Appendix B.1)
> > It would also be helpful to give an indication of what from Schulman et al. (2015) is being applied (i.e., the what the "stochastic computation approach" is).
> > As a result, I'm not convinced that the reconstruction game, absent modifications to the traditional object (e.g., length penalty), assumes a uniform prior of messages.
>
> Thank you for pointing it out.
>
> Apart from the standard VAE, we adopted the REINFORCE-like optimization but the "reward" (reconstruction term) is differentiable (w.r.t the receiver $\theta$).
> The differentiability of the "reward" hinders the direct application of REINFORCE.
> That is why, instead of Williams (1992), we mentioned Schulman et al. (2015), who proposed some sort of the generalization of REINFORCE.
> As you pointed out, our proof would become clearer by explicitly explaining how to utilize the result of Schulman et al. (2015).
>
> ## Revision for Comment 11
> To clarify the proof, we added two lemmas (Lemma 1; 2) in Appendix B.1 to support the proof description (The lemmas can be seen as a special case of Schulman (2015) and/or REINFORCE).
>
> ## Reviewer's Comment 12 (about Sec 3.1; Appendix B.1)
> > It seems like the $P(m)\sim\mathop{\mathbb{E}}\nolimits_{x\sim P_{obj}}[S(m|x)]$ should be the prior over messages.
>
> We understand your concern, but this is not the case.
> We hope that the above revision made our proof persuasive enough to regard the uniform distribution as the prior.
>
> Let us explain why the average sender is not the prior in the conventional EC signaling game, using some equations.
> First, let us decompose Eq. 9 as follows:
>
> $$
> \begin{aligned}
> \nabla_{\phi,\theta}\mathop{\underbrace{\mathop{\mathbb{E}}\nolimits_{P_{\textrm{obj}}(x),S_{\phi}(m|x)}[\log R_{\theta}(x|m)]}}\limits_{\text{conventional objective}}
> &\mathop{\underbrace{=}}\limits_{\text{proof target}}
> \nabla_{\phi,\theta}\mathop{\mathbb{E}}\nolimits_{P_{\textrm{obj}}(x),S_{\phi}(m|x)}[\log R_{\theta}(x|m)+\log P_{\textrm{unif}}^{\textrm{prior}}(m)]
> \\\\
> &=
> \mathop{\underbrace{\nabla_{\phi,\theta}\mathop{\mathbb{E}}\nolimits_{P_{\textrm{obj}}(x),S_{\phi}(m|x)}[\log R_{\theta}(x|m)]}}\limits_{\text{same as the left-hand side.}}
> +\mathop{\underbrace{\nabla_{\phi,\theta}\mathop{\mathbb{E}}\nolimits_{P_{\textrm{obj}}(x),S_{\phi}(m|x)}[\log P_{\textrm{unif}}^{\textrm{prior}}(m)].}}\limits_{\text{should be proven $=0.$}}
> \end{aligned}
> $$
>
> Based on this decomposition, what we have to prove essentially is: $\nabla_{\phi,\theta}\mathop{\mathbb{E}}\nolimits_{P_{\textrm{obj}}(x),S_{\phi}(m|x)}[\log P_{\textrm{unif}}^{\textrm{prior}}(m)]=0$.
> According to Lemma 1 (which is added in the revised version), it can be further transformed as follows:
>
> $$
> \begin{aligned}
> \nabla_{\phi,\theta}\mathop{\mathbb{E}}\nolimits_{P_{\textrm{obj}}(x),S_{\phi}(m|x)}[\log P_{\textrm{unif}}^{\textrm{prior}}(m)]
> &\mathop{\underbrace{=}}\limits_{\text{From Lemma 1.}}
> \mathop{\mathbb{E}}\nolimits_{P_{\textrm{obj}}(x),S_{\phi}(m|x)}[\nabla_{\phi,\theta}\log P_{\textrm{unif}}^{\textrm{prior}}(m)+(\log P_{\textrm{unif}}^{\textrm{prior}}(m))\nabla_{\phi,\theta}\log S_{\phi}(m|x)].
> \\\\
> &=
> \mathop{\underbrace{\mathop{\mathbb{E}}\nolimits_{P_{\textrm{obj}}(x),S_{\phi}(m|x)}[\nabla_{\phi,\theta}\log P_{\textrm{unif}}^{\textrm{prior}}(m)]}}\limits_{(\dagger)}
> +\mathop{\underbrace{\mathop{\mathbb{E}}\nolimits_{P_{\textrm{obj}}(x),S_{\phi}(m|x)}[(\log P_{\textrm{unif}}^{\textrm{prior}}(m))\nabla_{\phi,\theta}\log S_{\phi}(m|x)]}}\limits_{(\ddagger)}.
> \end{aligned}
> $$
>
> Here,
> - Term $(\dagger)$ would be $0$ if $\nabla_{\phi,\theta}\log P_{\textrm{unif}}^{\textrm{prior}}(m)=0$.
> - Term $(\ddagger)$ would be $0$ if $\log P_{\textrm{unif}}^{\textrm{prior}}(m)$ is constant w.r.t $m$ (See Lemma 2, which is added in the revised version).
>
> Thus, we should define $P_{\textrm{unif}}^{\textrm{prior}}(m)$ as a uniform distribution, i.e., $P_{\textrm{unif}}^{\textrm{prior}}(m)=\frac{1}{|\mathcal{M}|}$.
>
> Now, let us assume a prior distribution to be $S_{\phi}(m):=\mathop{\mathbb{E}}\nolimits_{x\sim P_{\textrm{obj}}}[S_{\phi}(m|x)]$ as you suggested.
> Then, what we have to check (prove/disprove) is whether $\nabla_{\phi,\theta}\mathop{\mathbb{E}}\nolimits_{P_{\textrm{obj}}(x),S_{\phi}(m|x)}[\log S_{\phi}(m)]=0$.
> But the gradient is not zero in general, since
> $$\mathop{\mathbb{E}}\nolimits_{P_{\textrm{obj}}(x),S_{\phi}(m|x)}[\log S_{\phi}(m)]=-\mathop{\mathbb{E}}\nolimits_{S_{\phi}(m)}[-\log S_{\phi}(m)]=-\mathcal{H}(S_{\phi}(M)).$$
>
> ## Reviewer's Comment 13 (about Sec 3.2)
> > Sec 3.2: what is a "heuristic variant of [a] VAE"?
>
> We just meant by that phrase that the conventional EC signaling game is similar to VAE.
> Indeed, it might be an awkward phrase as you imply.
> ## Revision for Comment 13
> We rephrased the phrase as follows:
> > (***Revision***) In this sense, the conventional signaling game is similar to VAE.

---

> > ### Author Response · Authors · 2023-11-17
> > **Response to Reviewer E7HT (Comment 14-19)**
> >
> > ## Reviewer's Comment 14 (about Sec 3.4)
> > > The very first paragraph of this section, I think, is glossing over critical question in the paper: what is the connection between the "prior" and the actual EC setup. I understand that the EC setup is analogous to a VAE, but what exactly is the analog of the VAE's prior?
> >
> > We appreciate your comment regarding what role the prior should play in EC setup.
> > For clafication, let us contrast the pov of previous work and the pov of ours.
> >
> > The pov of previous work:
> > - In the first place, they are not aware of the implicit prior distribution.
> > - Thus, the prior distribution is just an unknown, unexpected artefact.
> >
> > The pov of ours:
> > - In this paper, we consider the prior as the receiver's language model (LM) $P_{\theta}^{\textrm{prior}}(m)$.
> > - (Somewhat surprisingly, the idea of introducing LM into the EC signaling game has not been done before to the best of our knowledge, even though the idea per se seems quite natural.)
> >
> > ## Revision for Comment 14
> > We largely wrote the beginning of Sec 3.4 to clarify our position, as follows:
> > > (***Revision***) ... However, the signaling-game counterpart of the VAE's prior distribution has not been clear yet. In other words, as a sub-domain of linguistics, it is important to take a clear stance on which part of real-world communication it models. In this paper, we consider the prior distribution as the receiver's neural language model (LM) parametrized by $\theta$: ...
> >
> > Also, to make our positioning clearer, we added the following phrase in Sec 1 (Introduction):
> > > (***Revision***) Interestingly, the prior distribution can be seen as a language model, and thus the corresponding term can be seen as...
> >
> > ## Reviewer's Comment 15 (about Sec 3.4)
> > > I think the use of "approximately" is dangerous when trying to make theoretical claims; I understand that it isunavoidable in something as messy as EC, but it still needs to be accompanied by some justification in order tokeep the theoretical claims strong.
> >
> > We agree with your point.
> >
> > ## Revision for Comment 15
> > Just after Eq 15 in Sec 3.4, we added some more equations to imply a prior distribution "approximates" the average behavior of a sender, as follows:
> > > (***Revision***) ... It can be seen as an LM because it approximates the average behavior of a sender, i.e., the KL term in Eq 7, $\mathop{\mathbb{E}}\nolimits_{P_{\textrm{obj}}(x)}[D_{KL}(S_{\phi}(M|x)||P_{\theta}(M))]=-\mathop{\mathbb{E}}\nolimits_{P_{\textrm{obj}}(x)}[\mathcal{H}(S_{\phi}(M|x))]-\mathop{\mathbb{E}}\nolimits_{S_{\phi}(m)}[\log P_{\theta}(m)]$, is minimized w.r.t $\theta$ when $P_{\theta}(m)=S_{\phi}(m):=\mathop{\mathbb{E}}\nolimits_{P_{\textrm{obj}}(x)}[S_{\phi}(m|x)]$. ...
> >
> > ## Reviewer's Comment 16 (about Sec 1)
> > > "they are not" reproduced emergent lanuaguages has awkward phrasing
> >
> > ## Revision for Comment 16
> > We rephrased it as follows:
> > > (***Revision***) It has been reported that emergent languages do not give rise to these properties ...
> >
> > ## Reviewer's Comment 17 (about Sec 2)
> > > Right after Eq. (1), it should be $\log(|\mathcal{A}|-1)\geq 0$ in the case that $|\mathcal{A}|=2$.
> >
> > In fact we meant $|\mathcal{A}|\geq 3$  by `We assume $\mathcal{A}$ contains at least 2 symbols other than $\texttt{eos}$`.
> > But it was indeed an awkward phrasing.
> >
> > ## Revision for Comment 17
> > We rephrased it as a more direct one:
> > > (***Revision***) We assume $|\mathcal{A}|\geq 3$
> >
> > ## Reviewer's Comment 18 (about Sec 3.3)
> > > - what is $\mathcal{M}$ -- the set of all messages?
> > > - What is $\mathcal{A}$, again?
> >
> > Yes, their notations are the same as Sec 2.
> >
> > ## Revision for Comment 18
> > To clarify common mathematical notation throughout this paper, we made `Mathematical Notation` section at the beginning of Sec 2.
> >
> > ## Reviewer's Comment 19 (about Sec 3.3)
> > > Eq 15 limit notation here would be more appropriate
> >
> > Thank you for pointing it out.
> >
> > ## Revision for Comment 19
> > - We wrote Eq 15, using limit notation.
> > - (But note that the corresponding paragraph moved to Appendix B.3)

---

> > ### Comment · Reviewer_E7HT · 2023-11-21
> > **Revised evaluation in light of rebuttal**
> >
> > I believe that the rebuttal and revisions of the paper have addressed the weaknesses stated in my review and bolstered the strengths.
> > Specifically, I will change the following ratings in review:
> >
> > - Soundness: 2 -> 3
> > - Presentation: 2 -> 3
> > - Rating: 5 -> 8
> >
> > The reason that I am willing to change my rating significantly is that (1) I was already convinced of the importance of approaching EC theoretically before immediately jumping to empirical evaluation as well (expanded upon in my reply to Reviewer unmd) as the particular analysis of the signalling game as an VAE (I think it is an almost obvious analogy but one that has lacked substantial analysis in the extant literature).
> > (2) And my main concerns had to do with the clarity and the exact nature of one of the central claims (i.e., the "prior" of the signalling game/VAE).
> > Since the clarity issue was resolved adequately, I feel that the strength of the approach can shine through, and that the theoretical analyses can be of use to practitioners down the road.
> >
> > I acknowledge the weaknesses given by Reviewer TgUR, and I think they have been sufficiently ameliorated by the revisions, even if not completely resolved.
> > The weaknesses given by Reviewer unmd are not substantive and do not impact my evaluation significantly.

---

### Official Review · Reviewer_knBD · 2023-10-30

**Soundness:** 3 good
**Presentation:** 4 excellent
**Contribution:** 4 excellent
**Rating:** 8
**Confidence:** 3

**Summary:**

This paper introduces a new perspective on Lewis’s signaling game as beta-VAE and reformulates the game’s objective as ELBO. Based on this modification, it analyzes the influence of the implicit prior function on the properties of word lengths and segmentation of the emergent languages. It also shows that a learned prior distribution of the emergent languages can help evolve a language following Zipf’s law and Harris’s articulation scheme while the previous conventional objectives do not encourage meaningful segments.

**Strengths:**

1. The originality of this paper is good. The authors propose a generative point of view of the signaling game and analyze the possible causing factors of the current problems of the emerging less meaningful linguistic properties using the conventional objectives. This can provide a fresh study framework for emergent communication. The rigorous formalization and mathematical equations can integrate previous designs of regularizers and help with future objective design, offering a valuable contribution to the field.

2. The quality of the experiments and analysis is good. They compare different baselines controlling different priors of the objectives. The properties of word lengths, segments, and compositionality are carefully checked.

3. This paper is of good clarity. It is easy to follow the argument of this paper.

**Weaknesses:**

No obvious weaknesses.

**Questions:**

Based on the current formulation, it seems that the distractors on the receiver’s side are not considered. How would you incorporate the context of the distractors and their corresponding influences [1,2] into the prior design?

[1] Lazaridou, Angeliki, Alexander Peysakhovich, and Marco Baroni. "Multi-agent cooperation and the emergence of (natural) language." arXiv preprint arXiv:1612.07182 (2016).

[2] Evtimova, Katrina, et al. "Emergent communication in a multi-modal, multi-step referential game." arXiv preprint arXiv:1705.10369 (2017).

---

> ### Author Response · Authors · 2023-11-17
> **Response to Reviewer knBD**
>
> We greatly appreciate your positive feedback on our research!
>
> > Based on the current formulation, it seems that the distractors on the receiver’s side are not considered.
> > How would you incorporate the context of the distractors and their corresponding influences into the prior design?
>
> This is an important point. Thank you for highlighting it.
>
> Similar to Lewis's signaling (reconstruction) game, Lewis's discrimination (referential) game has also been a significant model in EC.
> Due to the generative nature of our storytelling, we have focused on the reconstruction game in this study.
> We plan to extend our study to include the discrimination game in our future work.

---

### Official Review · Reviewer_unmd · 2023-11-04

**Soundness:** 2 fair
**Presentation:** 1 poor
**Contribution:** 2 fair
**Rating:** 3
**Confidence:** 5

**Summary:**

The paper attempts to reframe the conventional Lewis's signaling game within the context of beta-VAE and ELBO, with a focus on the impact of prior distributions on emergent languages. The authors argue that selecting appropriate prior distributions can lead to the emergence of more natural language segments, while the conventional prior may hinder adherence to linguistic properties like Zipf's law of abbreviation (ZLA) and Harris's articulation scheme (HAS).

The weak points of this paper include:
(1) The paper is hard to read. The theoretical section includes symbols and equations without full explanation.
(2) The experiments are weak. The compared methods lack descriptions, and the performance improvement is not well explained.
(3) The studied problem lacks of enough audience.

**Strengths:**

1. The author well introduces the problem, which is well motivated.
2. The authors provide a deteailed proof in supplementary material.

**Weaknesses:**

(1) The paper is hard to read. The theoretical section includes symbols and equations without full explanation.
(2) The experiments are weak. The compared methods lack descriptions, and the performance improvement is not well explained.
(3) The studied problem lacks audience in the community.

**Questions:**

I suggest the authors address my concerns mentioned in Weaknesses.

---

> ### Author Response · Authors · 2023-11-17
> **Response to Reviewer unmd**
>
> Thank you for reviewing our paper.
>
> ## Reviewer's Comment 1
> > The paper is hard to read. The theoretical section includes symbols and equations without full explanation.
>
> Thank you for pointing out the need for clearer explanations in the theoretical section.
> Taking all the reviewers' comments into consideration, we have now added explanations for symbols and equations.
>
> If you have any other specific problems or comments, please do not hesitate to mention them.
>
> ## Reviewer's Comment 2
> > The experiments are weak. The compared methods lack descriptions, and the performance improvement is not well explained.
>
> We appreciate your feedback regarding the experimental section.
>
> We believe we have adequately addressed the important aspects of the theoretical contribution through mathematical proof. Regarding word length and segmentation, we are confident that the experimental results we have obtained support the main claims of our contribution.
>
> However, it is possible that we may have overlooked some points that the reviewer has noticed. If you believe there is a lack of experiments in terms of supporting specific hypotheses, we would greatly appreciate your feedback.
>
> ## Reviewer's Comment 3
> > The studied problem lacks of enough audience.
>
> We are afraid that you may be underestimating the growing interest in Emergent Communication (EC) and Representation Learning.
> Both of them are increasingly being recognized as important in recent years, as evidenced by the significant number of papers published in major machine leaning conferences.
> Our paper bridges these two areas by demonstrating how EC signaling game can be re-interpreted as representation learning in generative models, namely beta-VAE.

---

> ### Comment · Reviewer_E7HT · 2023-11-21
> **Studied problem does have an audience**
>
> I would explicitly disagree with Weakness (3) given by Reviewer unmd. I think theoretical analysis of emergent communication games is much needed in the field as a whole.  For the most part, EC works lean towards "engineering" approaches (e.g., tinkering with empirical evaluation) instead of theoretical analysis that allows for principled predictions before empirical evaluation (even if they turn out to be wrong).  This allows for iterative development of theoretical models as they produce predictions which are subsequently tested and point to revisions needed in the theoretical model.  For this reason, I definitely think that the overall direction of this paper is important.

---

### Meta-Review · Area_Chair_vbGg · 2023-12-08

**Metareview:**

This paper presents a new perspective on Lewis's signaling game, reinterpreting it through the lens of beta-VAE and reframing the game's objective as the Evidence Lower Bound (ELBO). Building upon this innovative approach, the study investigates the impact of the implicit prior function on word length properties and the segmentation of emergent languages. Moreover, it demonstrates that a learned prior distribution for emergent languages can facilitate the development of languages that adhere to Zipf's law and Harris's articulation scheme, which conventional objectives fail to encourage.

The paper makes commendable efforts in establishing a unified theoretical framework for addressing the EC (Emergent Communication) problem, and the author responses effectively address many concerns raised by the reviewers.

Nonetheless, a few issues still persist:

- The paper's contribution and novelty appear somewhat limited when compared to existing work in the field.
- The argument for the increased High Agreement Score (HAS) lacks foundational support. The authors have acknowledged this limitation, but it continues to raise concerns among reviewers regarding the overall contribution.
- The experiments are weak and could benefit from further elaboration and additional details.

Overall, while the paper presents an interesting perspective on the signaling game and offers valuable insights into the emergence of languages, addressing the above-mentioned concerns would enhance its impact and contribution to the field.

**Justification For Why Not Higher Score:**

A few issues still persist after rebuttal:

- The paper's contribution and novelty appear somewhat limited when compared to existing work in the field.
- The argument for the increased High Agreement Score (HAS) lacks foundational support. The authors have acknowledged this limitation, but it continues to raise concerns among reviewers regarding the overall contribution.
- The experiments are weak and could benefit from further elaboration and additional details.

**Justification For Why Not Lower Score:**

Established unified theoretical framework for addressing the EC (Emergent Communication) problem. Good results. Many concerns raised by the reviewers have been addressed after rebuttal.

---

### Decision · Program_Chairs · 2024-01-16

Accept (poster)